# Self-propelling and rolling of a sessile-motile aggregate of the bacterium *Caulobacter crescentus*

Yu Zeng 🄳 [1] & Bin Liu 🄳 [1✉]

Active dispersal of microorganisms is often attributed to the cells' motile organelles. However, much less is known about whether sessile cells can access such motility through aggregation with motile counterparts. Here, we show that the rosette aggregates of the bacterium *Caulobacter crescentus*, although predominantly sessile, can actively disperse through the flagellar motors of motile members. Comparisons in kinematics between the motile rosettes and solitary swimming cells indicate that the rosettes can be powered by as few as a single motor. We further reconstructed the 3D movements of the rosettes to reveal that their proximity to a solid-liquid interface promotes a wheel-like rolling, as powered by the flagellar torque. This rolling movement also features a sequence of sharp turns, a reorientation mechanism distinct from that of swimming cells. Overall, our study elucidates an unexplored regime of aggregation-based motility that can be widely applied to sessile-motile composites.

[1] Department of Physics, University of California, Merced, Merced, CA 95343, USA. ✉email: bliu27@ucmerced.edu

 1

Microbial dispersal in an aqueous environment can be active or passive, based on the mechanism that drives the dispersion[1–3]. In active dispersal, individual cells are equipped with motile organelles[4–7], such as flagella or pili, which enable swimming (in liquids) or gliding motilities (near a surface), respectively. In dense populations of motile cells, cell-to-cell interactions lead to collective dispersal, such as three-dimensional (3D) vortical flows in bulk fluids[8,9] and quasi-two-dimensional (2D) swarming[10,11] above a semisolid surface, e.g., an agar plate. Also, collective motility can be achieved through aggregation, maintained through intercellular coalescence[12,13], as larger dispersal units. Within these modes of collective dispersal, member cells contribute uniformly and isotropically to the entire group, resulting in an overall diffusive group motility. In passive dispersal, the dispersal units consist of sessile cells alone, which lack self-powering motile organelles[14] and can only be transported through environmental entrainment[15,16]. An example is biofilm sloughing initiated and transported by the surrounding flow[2,17], although such unidirectional transport is passive and non-responsive to environmental stimulation.

In addition to these uniformly distributed motile or sessile dispersing units, assemblies of microorganisms with distinct physiological properties have been widely recognized in pathogenicity and ecological processes[18,19]. However, the roles of such a heterogeneity in motilities is rather underexplored. As a paradigm of such heterogeneities in motilities, a binary aggregate can be formed between both sessile and motile cells due to, for instance, variant possessions of motile organelles associated with asynchronous development in life cycles. This sessile–motile aggregate leads to an intermediate dispersal regime where active and passive dispersals coexist within the same dispersal unit. In this intermediate regime, it is unclear whether the motile compartmentis capable of carrying the entire aggregate and allocating resources.

To investigate this potential mode of dispersal, we explored the motility in rosette aggregates of an aquatic bacterium[12]: *Caulobacter crescentus*. *Caulobacter* bacteria are widespread in soil, aqueous environments, within organisms, as well as in clinical systems (e.g., tap water, human blood and invertebrate guts[20,21]). Gaining a full understanding of *Caulobacter* motility thus impacts many ecological and medical applications. Among *Caulobacter* species, the non-pathogenic strain, *C. crescentus* (CB15), has been most studied, primarily for understanding its asymmetric division and dimorphic life cycle[22]. The life cycle of a *C. crescentus* begins as a curved rod-shaped motile cell (known as a swarmer). A motile cell is propelled by the rotation of a single helical flagellum, which is powered by a flagellar motor that can operate in both clockwise (CW) and counterclockwise (CCW) directions[20]. An alternation of the motor direction leads to either a reverse in swimming direction (CW-to-CCW switch) or a flick followed by a more random reorientation (CCW-to-CW switch)[23,24]. These forward–reverse–flick switches lead to a random-walk mechanism distinct from that of many peritrichous bacteria (e.g., *Escherichia coli*), which rely on run-and-tumble to swim and reorient[25]. This motile stage eventually ends with the swarmer cell shedding its flagellum and growing a stalk. Each stalk possesses a polysaccharide holdfast at the distal end, which allows for adhesion to solid surfaces or the holdfast from another cell[26,27]. The cell then elongates its body and becomes a predivisional cell, which later undergoes asymmetric division and gives rise to a stalked mother cell and a flagellated daughter cell.

*C. crescentus* rosettes are aggregates of sessile stalked cells and have been regularly observed under laboratory condition[20,26,28]. Rosettes are frequently studied when investigating the mechanisms of adhesion[26] and mechanical properties of stalks in *C. crescentus*[28]. However, its biological significance and motility are unclear. Though visual evidence is lacking, it has been speculated that the rosettes are formed by random collisions between stalked cells at similar stages of growth[20], which attach to a common core through holdfasts[26,27]. The potential sessile–motile coexistence in rosettes is primarily driven by the asymmetric cell division[29]: each newly yielded daughter cell grows a functional motor and a flagellum. A finite division time, typically over an hour[30], is expected to cause intermittent bimodular states of rosettes that consist of both sessile and motile members. In this study, we found that *C. crescentus* rosettes can indeed achieve active dispersal despite being predominantly composed of sessile cells. By comparing the kinematics of rosettes with that of solitary swarmer cells, we also illuminated the mechanism that underlies such an aggregation-based dispersal.

## Results

**Sessile–motile coexistence leads to active rosette motilities.** *C. crescentus* rosettes are popularly found in agitated culture media, with their sizes dependent on the initial concentration of viable cells[20]. Under our culture condition (see "Methods"), most rosettes were roughly spherical structures of 5–10 μm in diameter (Supplementary Fig. 1), containing tens of member cells ($N_s$ = 15–25). To demonstrate that sessile and motile cells indeed coexisted in the *C. crescentus* rosettes, we imaged them under a scanning electron microscope (SEM). As shown in a typical rosette image (Fig. 1a), sessile member cells undergoing different growth stages appended to each other through their stalks near the center of the rosette. Here, a single predivisional cell (colored in Fig. 1a) was found equipped with a flagellar filament, representing the motile compartment in this sessile–motile aggregate.

In order to determine whether these bimodular aggregates exploit their motile compartments for any long-term motility, we recorded the positions of individual rosettes over a long period (30 min on average), comparable to the lifespan of a solitary swarmer cell[31]. These rosettes were suspended in the growth medium confined between two coverslips (with a distance ~50 μm), and visualized under an inverted microscope equipped with 3D automatic tracking (see "Methods"). Rosettes were mostly detected near the bottom coverslip (with a cell-wall gap $d \lesssim 10$ μm).

Rosettes displaced over 50 μm in 10 min (Fig. 1b and Supplementary Movie 1), a much longer distance than can be achieved through Brownian motion alone (3–5 μm, based on the diffusivity in water of a colloidal particle[32]). Over a sufficiently long period of time ($\gtrsim$10 min), the trajectories of rosettes were often interrupted by noticeable pauses, during which rosettes appeared almost immotile (speed $u \lesssim 0.5$ μm/s). This coexistence of both motile and immotile regimes presumably resulted from the dynamics of its powering source during cell division. In the following, we only focused on this motile regime.

The typical rosette trace resembled a random walk, but consisted mostly of curved paths rather than linear segments found in solitary swarmer cells (Supplementary Fig. 2)[23,24]. In addition, rosettes moved at a ballistic speed ($u = 3.6 \pm 2.3$ μm/s; mean ± SD, $N = 71$ rosettes) much slower than that of individual swarmer cells ($u = 45 \pm 19$ μm/s; mean ± SD, $N = 119$ cells). Despite these differences in kinematics, the mean-squared displacement (MSD) curves of rosettes and swarmer cells showed very similar trends (Fig. 1c): both were characterized by a ballistic regime (with a power-law exponent ≈ 2) within a lag timescale $\Delta t \approx 1$ s and a diffusive regime (with an exponent ≈ 1) at larger timescales. In self-propelled particles, such a crossover lag time between ballistic and diffusive regimes is typically associated with the duration of self-propulsion, beyond which a reorientation event is likely to occur. In the case of a swarmer cell, this

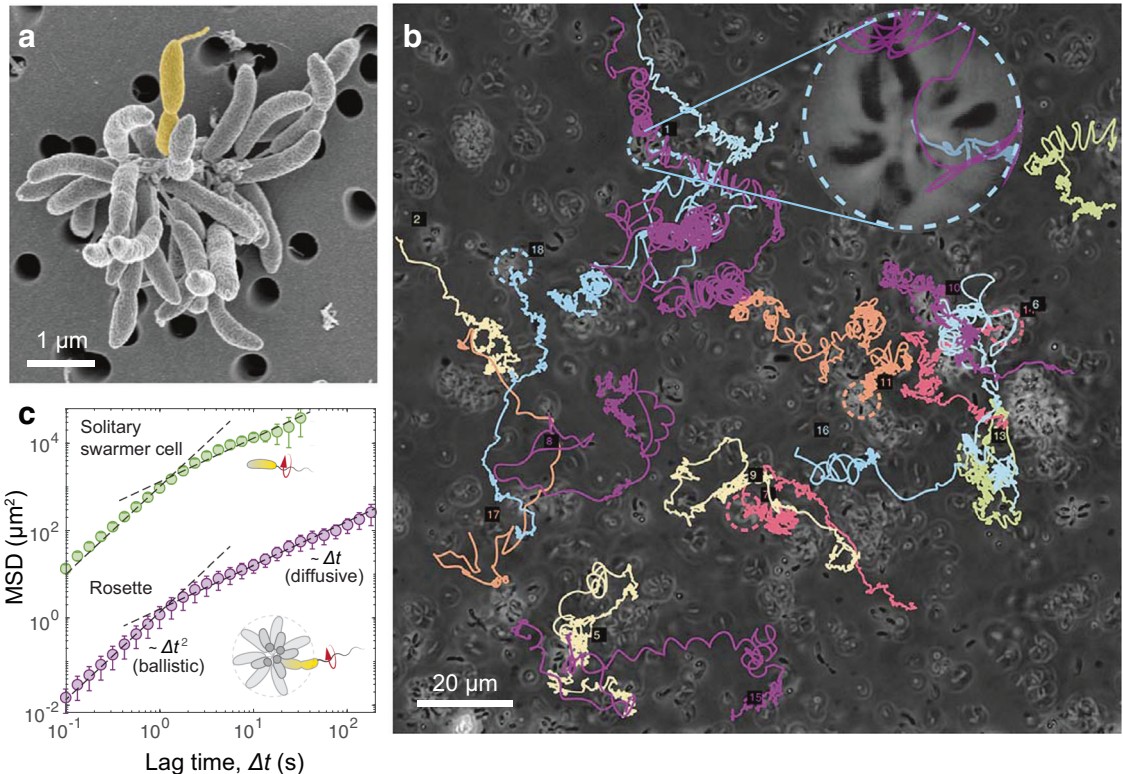

**Fig. 1 Dispersal of sessile–motile aggregates. a** Scanning electron microscopy of a typical *C. crescentus* rosette with numerous stalked cells and a flagellated predivisional cell (yellow). **b** Sample trajectories of 18 *C. crescentus* rosettes simultaneously recorded (in 10 min) via phase contrast microscopy. Those individual rosettes remaining in the view after 10 min are circled. A rosette is visible in the zoomed-in view (inset). **c** The mean-squared displacement (MSD) of nine rosettes (purple) demonstrates a crossover lag time ($\Delta t \sim 1$ s) demarcating the ballistic and diffusive regimes. The MSD of six individual solitary swarmer cells (green) shows a similar ballistic-diffusive transition, implying similarity in the temporal organization of power sources. Dashed lines, fits of the experimental data with both ballistic (MSD $\propto \Delta t^2$) and diffusive (MSD $\propto \Delta t$) activities. Error bars correspond to standard deviations. The difference in sample size between **b** and **c** is due to the fact that not all trajectories in **b** are sufficiently long for MSD calculations (see "Methods").

crossover timescale reflects the duration of a motor reversal cycle (CW-to-CCW-to-CW), which drives the run–reverse–flick process[23,24]. Similarly, propulsion with a single flagellar motor likely underlies the motility of rosettes despite the substantial size difference. In swarmer cells, sporadic reversals of the flagellar motor for cell reorientation likely underlie the finite ballistic regime (Fig. 1c). The same sporadic reversals may affect rosette maneuverability with reconfigured powering but similar temporal organization (see below).

**Rosettes' rotation in 3D**. Long-range displacements of rosettes were accompanied by rotational movements, which can be identified under the microscope by following the lateral displacements (i.e., movements in the *x-y* plane) of a rosette's member cells relative to its geometric center (Fig. 2a). These rotational movements were consistent with the spinning of the cell body of a swarmer cell, subjected to the torque produced by the flagellar motor[33]. The torque-free condition for a microscale swimmer (including all cell bodies and flagella) requires the rosette to rotate in a direction opposing that of any running motors. Given the same motor torque, the apparent larger payload for a rosette gave rise to its slower rotational speed.

Here, rosettes rotated at a much slower rate (with an estimated angular speed $\omega \sim 1$ rad/s) as compared to that of a solitary cell ($\omega \sim 200$–400 rad/s)[33]. Ignoring the potential boundary effects due to the nearby wall, the required torques for these rotations were well characterized by the viscous torque, i.e., $L = \sigma\omega$, where

$\sigma$ is the rotational drag coefficient. Considering a spheroid of a semi-major axis $R$ and an eccentricity $e$, we obtained the coefficient $\sigma = \frac{16e^3}{3\xi(e)}\pi\eta R^3$, where $\xi(e) = \frac{e}{1-e^2} - \frac{1}{2}\log\left(\frac{1+e}{1-e}\right)$[34]. Given rosettes in this study ($R = 2$–6 μm, $e = 0$; see Supplementary Fig. 1) and a solitary cell ($R = 1$ μm, $e = \sqrt{3^2-1}/3 = 0.94$ for a 3:1 aspect ratio[35]), the ratio between the computed rotational drag coefficients in these two cases gave $\sigma_{\text{rosette}}/\sigma_{\text{solitary}} \sim 9 \times 10 - 2 \times 10^3$. Combining with the ratio between the aforementioned angular speeds ($\omega_{\text{rosette}}/\omega_{\text{solitary}} \sim 1/400 - 1/200$), we concluded that the torque, a product of $\sigma$ and $\omega$, had the same order of magnitude for both the rosettes and the swarmer cells. In other words, the ratio between the torque on a rosette and that on a single-cell $L_{\text{rosette}}/L_{\text{swarmer}} = 0.2$–10. This scaling analysis thus implies that one flagellar motor can account for the observed rosette rotations, agreeing with our previous argument that a single motor can power the entire rosette. It is worth noting that we considered only the free-space situation in the above scaling analysis. However, including the hydrodynamic effects of a nearby wall did not alter our conclusion, as verified by a hydrodynamic simulation (Supplementary Method 1 and Supplementary Fig. 3).

To quantify the angular velocity $\omega$ in rosettes, we obtained the lateral movements of rosettes near the image plane through particle image velocimetry (PIV) and fitted them with the velocity field due to a 3D rigid-body rotation (Fig. 2b, c). Here, the finite focal depth, differentiating the actual velocity fields from that in a pure 2D plane, was used to reconstruct the 3D angular velocities

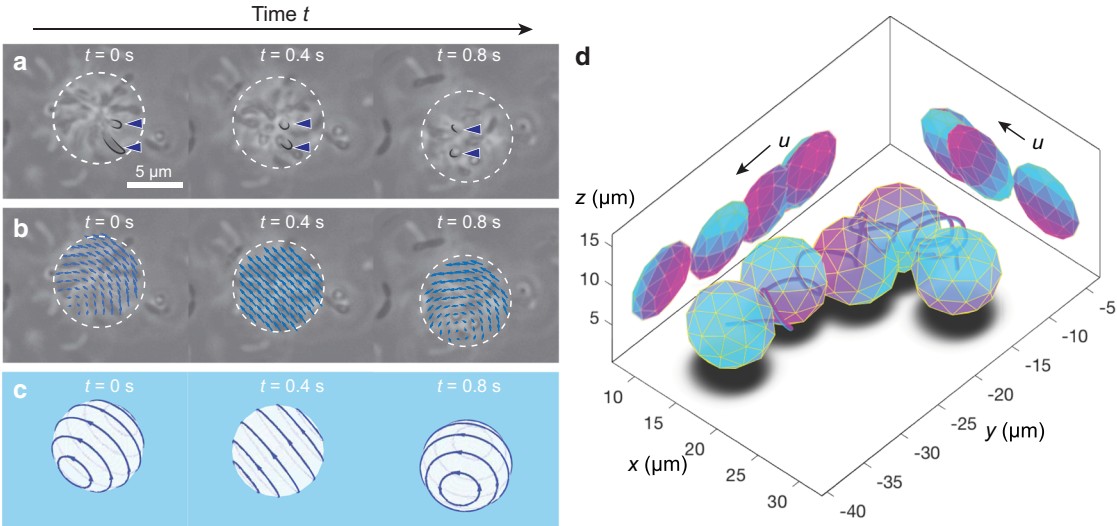

**Fig. 2 Self-rotation of a *C. crescentus* rosette. a** The lateral displacement of member cells of a rosette relative to its geometric center suggests rotational movement. Here, two member cells of a rosette (arrowheads) are imaged undergoing such relative displacement. **b** These relative displacements are best fitted by an in-plane velocity field due to a rigid-body rotation (Supplementary Methods 2–4 and Supplementary Figs. 4–7). **c** Corresponding 3D rotational movements at $t = 0$ s, $t = 0.4$ s, and $t = 0.8$ s demonstrate the time-dependent rotation axis. **d** Snapshots (spheres) of one reconstructed rosette are shown along the rosette's trajectory (curved line). Two orthogonal views projected on the side walls show the dynamic orientation of this rosette during translation.

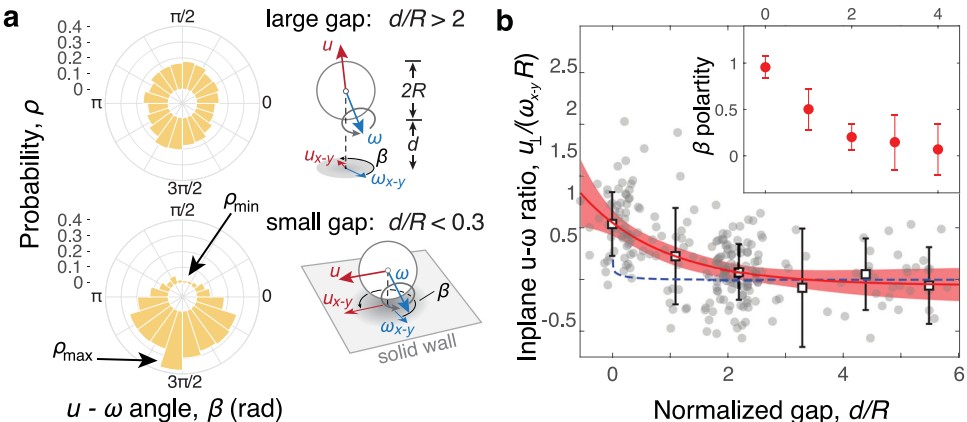

**Fig. 3 Rolling motility in *C. crescentus* rosettes. a** When there is a relatively large gap between the rosette and the solid wall ($d/R > 2$), the angle $\beta$ between the rotation ($\omega_{x-y}$) and translation ($u_{x-y}$) axes in the lateral plane (inset) shows no substantial bias between 0 and $2\pi$, suggesting complex rosette shapes and alignments of the flagellar axes. At small gaps ($d/R < 0.3$), $\beta$ is more polarized around $3\pi/2$, indicating a rolling movement. **b** Rotation–translation coupling is characterized by a slipping ratio, the ratio between the linear rotation speed $\omega_{x-y}R$ and the component of translational velocity $u_\perp$ perpendicular to the rotation axis. The observed slipping ratios, sampled over 1-s intervals (gray filled circles; means and standard deviations appear as squares and error bars, respectively; $N = 5$ rosettes), are plotted against the gaps between the rosette and the solid wall ($d/R$). The red line represents exponential fitting, with 95% confidence bounds in shade. Note that the observed ratios are higher than that expected from pure hydrodynamic interactions (dashed line) for $d/R < 1$, as further confirmed by the results from both linear regression and analysis of variance (ANOVA) (Supplementary Fig. 10 and Supplementary Method 7). The polarity of the probability function $\rho(\beta)$, i.e., $P = 2(\rho_{max} - \rho_{min})/(\rho_{max} + \rho_{min})$, is also shown as a function of $d/R$ in the inset (Supplementary Fig. 8 and Supplementary Method 5). The error bars show 95% confidence bounds.

(Supplementary Method 2, Supplementary Fig. 4 and Supplementary Movie 2). In addition, our use of phase-contrast optics rendered images that were sensitive to the distance of the rosette from the focal plane (Supplementary Methods 3 and 4 and Supplementary Figs. 5–7). Such axial dependencies were utilized for simultaneous reconstruction of rosettes' 3D translational movements, yielding more insight into the mechanism underlying the hybrid motility in this sessile–motile aggregate (Fig. 2d and Supplementary Movies 3 and 4).

**Solid surface promotes rosettes' rolling**. To determine how whole-rosette rotation contributed to long-range motility (and

ultimately cell dispersal), we characterized the translation–rotation coupling using an angle $\beta$, defined as the angle between the translation direction and the rotation axis in the plane along the bottom substrate, here, the $x$–$y$ plane (Fig. 3a, inset). When a rosette (with radius $R$) was sufficiently far from the bottom surface (with $d/R > 2$), the probability distribution $\rho(\beta)$ was slightly skewed toward $\beta \approx 3\pi/2$, if not uniform between 0 and $2\pi$ (Fig. 3a, upper panel). We attributed this rather uniform distribution to the complex rosette shape and the alignments of the flagellar axes, which did not necessarily point toward the rosette center. Such a complex configuration gave rise to a rich variety of angles between the rotation and translation axes. Moreover, even for a fixed angle

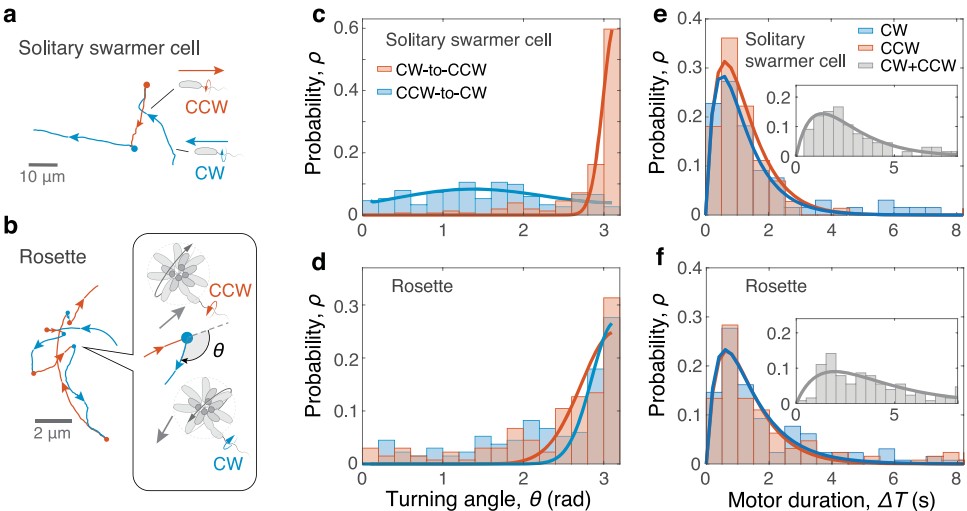

**Fig. 4 Comparison of reorientation mechanisms in solitary cells and rosettes. a** Trajectories for a solitary *C. crescentus* swarmer cell and **b** a rosette exhibit distinct geometries during motor switches. Motor switches appear as solid circles (each circle's color reflects the directionality following the switches). The turning angle $\theta$ indicates the corresponding change of translation direction (inset). **c** Turning of a solitary swarmer cell is demarcated by the direction of motor switch, with directional reversal ($\theta \approx \pi$) and flick ($\theta \approx \pi/2$) corresponding to CW-to-CCW and CCW-to-CW motor switches, respectively. **d** Rosettes generally exhibited sharp turns with $\theta$ concentrated near $\pi$ regardless of the direction of motor switch, in contrast to solitary cells. Solid lines correspond to curve fitting by multiple Gaussian distributions. **e** Durations of the flagellar motor in CW and CCW states are similar between solitary swarmer cells and **f** rosettes. Insets show the distributions of the restoring time of the flagellar motor (based on consecutive CW and CCW states), which characterizes the timescale of reorientation (Fig. 1c). Solid lines show curve fitting by multiple exponential functions. For **c–f**, the sampled numbers of motor switches are $N = 255$ and $N = 143$ for solitary cells and rosettes, respectively.

between these two vectors, the projection of this angle onto the $x$–$y$ plane is arbitrary for a rosette swimming in the bulk, subjected to the orientation of the plane formed by these two vectors. However, when the rosette was sufficiently close to the bottom surface (with $d/R < 0.3$), the probability distribution of $\beta$ was more polarized, centered around $3\pi/2$ (Fig. 3a, lower panel), indicating that the translation direction was more restricted to the direction perpendicular to the rotation axis. This probability distribution $\rho(\beta)$ can also be characterized for a range of gap $d$ (Supplementary Fig. 8 and Supplementary Method 5) by its polarity, i.e., $P = 2(\rho_{\max} - \rho_{\min})/(\rho_{\max} + \rho_{\min})$. Such a polarity $P$ decreased with increasing $d$, as shown in Fig. 3b (inset), consistent with a rolling movement induced by the bottom surface.

To quantify this rolling behavior, we measured the slipping ratio $Q$, defined by the observed linear displacement of a rosette normalized by that expected under the ideal no-slip situation, $Q = u_{\perp}/(\omega_{x-y}R)$. Here, $\omega_{x-y}$ is the in-plane angular speed and $u_{\perp}$ is the component of the in-plane linear velocity $u_{x-y}$ perpendicular to the rotation axis (Fig. 3a, insets). This slipping ratio became $Q = 1$ for no-slip rolling and $Q = 0$ for rotating with full slip. For rosettes near the bottom surface (with $d/R < 0.3$), we found that $Q \approx 0.5$ (Fig. 3b). To assess possible contributions from hydrodynamic interactions between the rosette and the bottom surface[36,37], we modeled a smooth sphere rotating in a Stokes fluid under external torque (Supplementary Method 6, Supplementary Fig. 9 and Supplementary Table 1) using a boundary integral simulation[38,39]. This modeling predicted a maximal slipping ratio ($Q_{\max} \approx 0.2$) still lower than that observed experimentally (Fig. 3b). The more no-slip-like rolling in rosettes thus likely relied on a more sophisticated rosette-wall interaction, for instance an interaction arising from the transient cell adhesion to the glass substrate.

**Motor reversals result in reorientations of rolling rosettes**. During rolling, a rosette's direction of translation depends on the bidirectional rotation state of the flagellar motor: CW or CCW as viewed from the exterior of the cell wall. In our experiment, the

state of a rosette's motor was conveniently revealed by the direction of circulation of its curved trajectory; the chirality in flagellar propulsion yielded CCW circulation for a CW motor and CW circulation for a CCW motor[36]. We should note that here we assumed that the flagellar axis was along the radial direction of the rosette (see Movies S5 and S6 for cases with more general flagellar alignments, e.g., along the orthoradial direction).

When a motor switched direction, the rosette reversed its direction of translation and almost retraced its previous path (Fig. 4 and Supplementary Movie 7). This forward–backward reorientation is distinct from the run–reverse–flick motion of solitary swarmer cells (Fig. 4a, b). Accordingly, the turning angle $\theta$ in rosettes concentrates near $\pi$ regardless of the type of motor switch (CW-to-CCW or CCW-to-CW), whereas $\theta$ in swarmer cells is sensitive to the state of the flagellar motor (Fig. 4c, d and Supplementary Fig. 11). This difference can be explained by considering the associated power configurations and movement mechanisms. For a swarmer cell, each flick reorientation, which corresponds to a CCW-to-CW motor switch and a realignment of the flagellar axis (Fig. 4a), is caused by the elastic buckling of the flagellar hook when flagellar thrust switches from pulling to pushing[23,24]. In contrast, the unanimous sharp turning of a rosette is an indicator that it is not propelled by a force parallel to the axis of the flagellar motor. Otherwise, the rosette's reorientation would be subject to a hook buckling during a CCW-to-CW motor switch, resulting in a flick similar to the solitary cell case. We thus speculated that the axis of rosette rotation was aligned with the direction of the flagellar torque, with its direction of translation perpendicular to its motor axis (Fig. 4b, inset).

We further examined the duration of the two flagellar motor modes in motor-switching events. The duration of flagellar motor activity ($\Delta T$) between two consecutive switches exhibited similar frequency distributions for rosettes and solitary cells, with negligible variation due to different motor directions (Fig. 4e, f). In both cases, the probability of $\Delta T$ peaked at $\Delta T \approx 1$ s, with rosettes showing heavier tails than swarmer cells (Supplementary Fig. 12). The amount of time necessary to restore the flagellar

motor to its previous state (the sum of durations of a CW state and a CCW state) was similar between solitary cells and rosettes, with peaks at $\Delta T \approx 2$ s (Fig. 4e, f, insets). This restoring time ($\Delta T \sim 1$ s) characterizes the reorientation timescales of both solitary cells and rosettes, as indicated by the match of the crossover times between the ballistic and diffusive regimes revealed by our MSD analyses (Fig. 1c). This similarity in motor behaviors further indicates that rosettes can be each powered by a single flagellated member cell.

## Discussion

In total, our experiments and analyses revealed that collective motility can arise from the aggregation between sessile and motile compartments, whose mechanism is distinct from all previously reported microorganism motilities.

The similarities between *C. crescentus* rosettes and solitary swarmer cells in their diffusive behaviors (Fig. 1c) and motor-switch statistics (Fig. 4c) unanimously indicate a likelihood that rosettes employed a single flagellar motor for active dispersal. This single-motor priority may potentially be limited by the scope of this workfor covering only motile rosettes and their movements near the bottom surface. For instance, it is possible for a rosette to have multiple flagella with their axes opposing each other, resulting in a potential offsetting of propulsion and therefore temporary immotility between motor switches. Such multiflagellation event should yield a fluctuation of rosettes' movements at a timescale of motor switches ($\Delta T \sim 1$ s), which was, however, not noticeable within our experimental period of time (>1 min). Meanwhile, the proximity to a solid surface may potentially impede the functionality of those motors facing the wall, leading to fewer effective motors for propulsion. Such fluctuations in the number and location of effective motors conflict with the persistent rotational movements observed in this study. The size of the rosettes (see Supplementary Fig. 1 for the size distribution) may also limit the number of available flagella given the relatively small fraction of the predivisional stage in each cell's life cycle. Based on an ideal statistical model and a known cell division timescale, $\Delta t_d \sim 10^2$ min[30,40], we found that, for rosettes in this study ($R = 4.0 \pm 1.2$ μm; mean ± SD; $N_s \lesssim 25$), the probability of having more than one active flagellum is as low as 3% (Supplementary Methods 9 and 10 and Supplementary Fig. 13). This time-dependent flagellation thus provides a candidate mechanism for the dominating single-motor powering of rosettes, as observed in this study.

The curved trajectories of rosettes were consistent with a hydrodynamic effect induced by the nearby solid surface[7], which was further demonstrated by a hydrodynamic model that simplified the rosette as a spherical object propelled by a rotating helical filament aligned in an arbitrary direction (Supplementary Fig. 2 and Supplementary Movies 5 and 6). While the transient behaviors of the simulated trajectories resembled a similar stochasticity and geometry (Fig. 1b), these simulations all converged to close orbiting of rosettes, different from their actual diffusive movements. It is plausible that the relatively more irregular shape of the actual rosettes, as compared to a smooth sphere, prevented them from approaching these periodic circulating states. This complex rosette geometry, together with its potential deformation upon a steric interaction with the wall, needs to be considered in greater detail to understand the full rosette kinematics.

Although the flagellar motor statistics (Fig. 4) suggested similar motor activities in both rosettes and solitary swarmer cells, the rosettes showed higher probabilities for relatively longer motor durations (Supplementary Fig. 12). These longer durations can result from potentially miss-counted intermediate motor reversals associated with indiscernible rosette displacements, due to, e.g.,

an extremely short motor duration or a relatively slow translational speed. Also, the potential mechanosensing of the rosette configuration by the predivisional cell and its variation in physiology during the generation of a flagellum may contribute to such a subtle difference in motor behaviors.

Through reconfigured powering and locomotor mechanisms, an aggregate like *C. crescentus* rosettes enables active transport of otherwise stationary cells. While unable to generate power for motilities, those stationary members ultimately determine the morphology of a rosette. Such spherical morphology contributes to the rosettes' unique rolling and reorientation movements and thus their colonization to a liquid-solid interface. Moreover, as individual members serve different roles in their life cycle (reproduction vs. dispersal), such a motility creates a synergy between both active motility and reproductivity, potentially increasing the fitness gain at the whole colony level. From an evolutionary perspective, a heterogeneous, multi-celled colony represents an intermediate stage between unicellular and multicellular forms. The adaptive value of such a heterogeneous *C. crescentus* rosette can thus be more systematically compared with that of a multi-celled colony with no intercellular differentiation (e.g., choanoflagellates[41,42]) to understand the evolutionary significance of such an intermediate stage.

Microbial aggregation exists in diverse forms, often as heterogeneous communities in nature[43]. The aggregation-based motility described here offers a spectrum of implications for understanding the potential diversities in microbial motility. For example, aggregation-based dispersal may facilitate pathogen transmission. In pathogenesis, many infectious microorganisms known to disperse passively[44,45] can disperse more rapidly if aggregated with motile cells, as evident by this study. Additionally, the aggregation between microorganisms and nonliving particles may also exhibit active motility and enhance the transport and exchange of environmental particles (e.g., a soil–microbe complex[46]). Following the architecture of a sessile–motile aggregate presented here, an aggregation-based design, with modularized motor and structural components, can be integrated with the existing microrobotic systems[47] for self-propelled and boundary-associated transport.

## Methods

**Swarmer cell and rosette preparation**. Wild-type *C. crescentus* (ATCC CB15) was cultured in peptone-yeast extract (PYE) growth media at 30 °C. A single colony from an agar plate was inoculated and incubated in a flask over a shaker (at 30 °C and 50 r.p.m.) for 8 h, before being transferred to a Petri dish and incubated (at 30 °C and 50 r.p.m.) overnight. Swarmer cells were obtained using a plate releasing technique[40]. The above Petri dish was thoroughly rinsed with deionized water to keep only stalk cells that attached to the bottom of the Petri dish. A small amount of growth media was applied to immerse the bottom of the Petri dish. After a 5-min wait, swarmer cells of similar ages (<5 min) were pipetted from the Petri dish and sealed between a coverslip and a glass slide (using vacuum grease) for microscope observations.

Two methods were used to obtain rosettes in this study. (1) We added 2 μl of the medium containing swarmer cells into the growth medium (~100 ml) in a flask, which was then incubated within a shaker (50 r.p.m.) at 30 °C for 18–20 h. (2) We prepared *C. crescentus* culture on an agar plate[20] and transferred ~0.1 μl of the culture into ~100 ml growth medium, which was then incubated within a shaker (50 r.p.m.) at 30 °C for 18–20 h. Rosettes of typically 5–10 μm in diameter were generated under both culturing conditions. For experimental filming, the culture was diluted by 2–3-fold, and a drop of the sample (~20 μl) was sealed between two coverslips (22 mm × 22 mm and 24 mm × 50 mm) using vacuum grease (separation ~ 50 μm).

**Swarmer cell and rosette imaging**. Filming was conducted using an inverted phase-contrast microscope (Nikon Eclipse Ti, ×100 oil-immersion objective). The sample was held on a three-axis piezo stage (Physik Instrument P-545.3D7, travel range of 70 × 70 × 50 μm) mounted on a motorized X–Y sub-stage (Prior Scientific, Inc., travel range of 100 × 76 mm), controlled via USB (Phidgets bipolar controllers and National Instruments NI USB-6211). Video frames were captured using a CCD camera (Allied Vision Technology PIKE F032B, 640 × 480 pixel at 208 fps). For *C. crescentus* swarmers, the microscope stages were programmed to move

along with an individual cell in 3D to maximize the tracking time[33]. The position of each cell was thus obtained from the position of the microscope stage. For *C. crescentus* rosettes, their relatively slow movements were followed by manually adjusting the microscope stage in 3D to avoid mistracking due to variant appearances. Rosette images were subjected to a zeroth-order Bessel function and a Gaussian filter such that each rosette appeared as a simply connected object (same as what we used for axial position reconstruction, see Supplementary Method 4). The 2D position of a rosette in each frame was thus determined by the center of the above filtered object.

**Scanning electron microscopy**. Culture medium with *C. crescentus* rosettes was first fixed using 3% glutaraldehyde in 0.1 M phosphate buffer (pH 7.2) for 1 h. The sample was then filtered using polycarbonate filter (0.4 μm; Whatman Nuclepore). After three washes in buffer, the sample was dehydrated in an ethanol gradient (30%, 50%, 75%, 90%, and 100%, 5 min each). The sample was then critical-point dried and sputter-coated with gold (5-nm-thick). Samples were then mounted on SEM stubs and examined with a ZEISS Gemini 500. Given partially damaged flagella in preparation (Supplementary Fig. 14) and the inaccessibility of a bottom view, this protocol was not used for counting the total number of flagella but for confirming the presence of flagella in cultured rosettes.

**Rosette morphology**. To measure rosette size, we recorded *z*-stack images of each rosette by scanning the sample along the *z*-axis using the piezo stage (at 100 step/s in 0.4 μm increments). A projection of the entire *z*-stack on the *x*–*y* plane was obtained by ImageJ and converted into a binary image. The rosette radius $R$ was given by the minimum bounding circle of the projection. The number of member cells in each rosette was counted from 3D reconstructions of *z*-stack images using ImageJ.

**Swarmer cell and rosette dispersal**. The motility of a rosette near a solid surface and that of a swarmer were both characterized by their 2D mean-squared displacement (MSD) in the *x*–*y* plane, i.e., $MSD(\Delta t) = <|\mathbf{r}(t + \Delta t) - \mathbf{r}(t)|^2>_t$, where $\mathbf{r} = (x, y)$ is the 2D displacement in the *x*–*y* plane, $\Delta t$ is the lag time, and the notation $< \cdot >_t$ denotes an average over time $t$. In the swarmer cell case, due to the relatively thin sample thickness ($\lesssim 50$ μm) and the relatively high swimming speed ~50 μm/s, the axial (*z*) movements were geometrically restricted to be quasi-2D. In addition, the *z* range of all swarmer cells that we recorded (for over 10 s) was within 10 μm, presumably subjected to a hydrodynamic attraction from coverslip surfaces[48]. We therefore concluded that the 2D MSD also characterizes reasonably well the diffusive behaviors of the swarmer cells. A crossover lag time $\Delta t_c$ that demarcates the ballistic and diffusive regimes was calculated by the intersection of a quadratic and a linear fit of the MSD vs. lag time curve. To avoid the potential bias from short rosette tracks, rosettes that stayed active within the view for over a minute were used for MSD calculation (Fig. 1c). Not all 18 rosettes in Fig. 1b satisfied this criterion, which led to a discrepancy in numbers of tracks between Fig. 1b and c.

**Rosette kinematics**. Rosettes were treated as rigid bodies due to the high stiffness of stalks and holdfast adhesion[28]. The 3D trajectories of rosettes were reconstructed by integrating the offset of the rosette from the image center, instantaneous stage position, and the estimated focal depth of the microscope (Supplementary Method 4). Rotation movements were reconstructed by fitting a 3D rigid-rotating model to a quasi-2D velocity field (obtained from the PIV) of the rosette near the focal plane (Supplementary Methods 2 and 3).

**Statistics and reproducibility**. Experimental data in this study were collected through multiple independent trials and their values were characterized by mean ± standard deviations, provided together with the corresponding sample sizes. Statistical tests were performed by standard regression (linear and nonlinear) algorithms coded in MatLab and pairwise comparisons using one-way analysis of variance (ANOVA) coded in R[49]. These results were considered significant only if $p < 0.05$.

**Reporting summary**. Further information on research design is available in the Nature Research Reporting Summary linked to this article.

## Data availability
Source data are provided with this paper[50]. Other data that support the findings of this study are shown in Supplementary Information or available from the corresponding author upon reasonable request.

## Code availability
The MatLab code for reconstructing the 3D positions and movements of rosettes is available on Github: 2D lateral position[51] (https://doi.org/10.5281/zenodo.4008579) 3D rotation, and axial position[52] (https://doi.org/10.5281/zenodo.4008588). The MatLab code for the hydrodynamic simulation of a self-propelling rosette is also available on Github[53] (https://doi.org/10.5281/zenodo.4008573). Raw images and position data from the microscope are

collected by a custom code programmed in Objective C, which is available from the corresponding author upon reasonable request.

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

## Acknowledgements

The authors thank Robijn Bruinsma, Itai Cohen, Ajay Gopinathan, and Jeremias Gonzalez for useful discussions. We also thank Joanna Valenzuela, Joseph Andrade, Taniya Badwal, Gloria Ligunas, and Sebastien Darbouze for assisting with experiments and collecting preliminary data. This work was supported by National Science Foundation Grant CBET-1706511. B.L. is grateful for support from NSF-CREST: Center for Cellular and Bio-molecular Machines (CCBM) at UC Merced (HRD-1547848) and Hellman Foundation.

## Author contributions

Y.Z. and B.L. designed and performed the research, analyzed the data, and wrote the manuscript.

## Competing interests

The authors declare no competing interests.
