## [Peer Review File · Communications Biology]

Reviewers' comments:

Reviewer #1 (Remarks to the Author):

In their manuscript Zeng and Liu report on the active dispersal and motility of rosette aggregates of *Caulobacter crescentus*. They study the 3D rotational and translational motion of such bacterial aggregates, which consist of both motile and sessile, i.e. non-motile, constituents. In particular, the authors show that the torque generated by a flagellar motor results in a wheel-like rolling motion of aggregates in the vicinity of a solid/liquid interface.

The manuscript, in general, is very well written and the ideas are clearly presented. The experimental approach is described concisely and the results are comprehensively elaborated in the main text, the figures as well as in the supplementary material. In particular, the figures provide clear and visually very appealing representations of experimental data. In addition to the high quality of the manuscript, also the system under investigation is scientifically very interesting. Much has been learned over the past decades on individual motile cells, e.g. spermatozoa, *E. coli* and *B. subtilis*, as well as cellular aggregates and populations of motile cells. However, the motility of multi-cellular entities exhibiting bimodal states of their cellular constituents still remains elusive to date.

In my opinion the work does satisfy the publication criteria of *Communications Biology*. The paper significantly advances the field of motile active matter and it is also of great interest to a general audience. The research has been conducted well and the results are - to the best of my knowledge - entirely novel.

I'd like to raise a few important points that the authors might want to address in their revised manuscript:

- a) Both individual cells and aggregates exhibit a transition from ballistic to diffusive at around 1 second as displayed in the exemplary MSD curves. What is the physiological meaning of this time scale? Can this be related to a tumble time or a rate at which reorientations occur? The authors should also elaborate more on the propulsion characteristics of individual *C. crescentus* cell (i.e. single flagellum, multiple flagella and bundling like in *E. coli* etc.). The authors speculate that the fact that this transition time is conserved for the aggregates might indicate a common underlying mechanism of propulsion at work. What is known about this mechanism? On page 4 they discuss "run-reverse-flick" motion and some of the above questions. I think this should be introduced earlier in the manuscript since not many readers might be familiar with this type of motility.
- b) Rotational degrees of freedom: The authors neglect any boundary effects on the rotational motion and, in particular, the torque calculations. What about adhesion and friction of sessile cells at the glass surface? Are the aggregates hovering above the the glass slide or is there any evidence of direct contact at the solid/liquid interface? Provided there is friction, one might need to consider more than just one motile cell being involved in rotation la motion.
- c) Can the size of an aggregate be adjusted experimentally - e.g. via culture media and/or cultivation procedures? In other words, which parameters determine the number of cellular constituents forming one aggregate?
- d) Motility mutants: Is there a way to test the hypothesis that an aggregate is powered by precisely one flagellar motor? Are motility mutants of *C. crescentus* with distinct motility characteristics available, which could then be used in analogous experiments?
- e) Collective effects: Have the authors performed experiments at high aggregate densities? How do two (rotating) aggregates interact and do they form coupled states of motility (see, e.g., Volvox colonies in the vicinity of a glass surface)?

For the reasons outlined above, I strongly support publication of the manuscript in *Communications Biology*, after my questions and remaining points of criticism have been adequately addressed.

Reviewer #2 (Remarks to the Author):

Zheng and Liu are reporting an interesting observation that the motion of *Caulobacter* rosettes is powered by a flagellated cell. One caveat of the study appears to be the assumption that motile rosettes have only one flagellated cell. The data that supports this assumption is not sufficient (only one EM image), yet the authors find that the duration of flagellar motor activity between two consecutive switches exhibits similar frequency distributions for rosettes and solitary cells. This indicates that either there is a mechanism that allows only one flagellated cell per rosette or somehow the authors have developed a bias in their data collection towards rosettes that only have one flagellated cell. My specific comments are below:

1. On an average, how many motile cells are found in a rosette? What is the probability of observing only one motile cell in a rosette? Since the authors presumably have multiple EM images collected without a selection bias, they can provide the statistics.
2. How does positioning of multiple motile cells impact motion of the rosette? Are there rosettes where no motility is observed despite the presence of motile cells? Does positioning of motile cells at opposite ends of the rosette cancel out/decrease net motility or rolling of the rosette?
3. The authors provide a calculation suggesting that a single motor can power a rosette's rotation but they don't tell us the prevalence of this scenario. One way to test it might be to fluorescently label the flagellum and count the number of flagellated cells in motile rosettes.
4. Could the authors provide videos of rosettes near and far from a solid surface and compare their rolling? This might make their statement that 'solid surfaces promote rosettes rolling' stronger.

Reviewer #3 (Remarks to the Author):

The manuscript "Self-propelling and rolling of a sessile-motile bacterial aggregate" describes self-propulsive motion of aggregates of active and passive cells of *C. crescentus* which forms a rosette like morphology. Specifically, the authors have pointed out an interesting observation that the motility of the aggregate is basically powered by a single flagella motor. The authors have compared the statistics of a solitary cell with a sessile-motile aggregate and found that the rosette motion follows intermediate values of diffusivities and rotation. The effect of solid boundary and motor reversal effects are also demonstrated. How a passive cluster of cells can show long range active motion by the presence of motile units is the key result of the present work. However, there are few points which need to be addressed before the manuscript can be published.

1. The underlying dynamical process and mechanism of the self-propulsive behavior needs to be explored in detail. How a single motor can bear such a load of sessile aggregate to rotate and roll is not very clear. The author should provide a proper mechanistic explanation of the origin of such motion to move a big unit of aggregated cell.
2. How significant is the roll of hydrodynamics interaction with the surrounding medium for the active dispersal?

3. For a dense system of cells, is there any suppression of the self-propulsive rotational motion? Or, bigger aggregates of sessile and motile cells may emerge in dense system. Specifically, with the variation of cell number density how the morphology and spatiotemporal dynamics changes?

Reviewer #4 (Remarks to the Author):

Contribution

The authors study the motility of an assembly of motile and sessile cells, the rosettes of *C. crescentus*. They use SEM and phase contrast microscopy to visualize the rosettes and analyze their motion. Using visualization, a 3D reconstruction of the motion, torque estimations and trajectory analysis, they deduce that the rosettes are powered by a single flagellated cell that creates a tumbling motion when the rosettes are close to the surface.

Summary

Overall, the manuscript is well written and the data is supporting the claims. The experimental methods are thoroughly executed and novel, in particular the 3D tracking of the rosette motion. Three major (but nonetheless easily accessible to the authors) points would help perfect this manuscript to make it worthy of a publication in *Communications Biology*:

1. More context about the *C. crescentus* rosette should be provided necessary to better frame the significance of the study, its contribution to the field. In particular, it is currently unclear how much this system has been described and studied in the past (not clear by looking at the references). More on the emergence (e. g. conditions of emergence, prevalence in natural settings) of these rosettes would also be helpful. The reviewer therefore recommends to remodel the third paragraph of the introduction to better put into context the findings of this biophysical study for the biology oriented readership of the journal.

2. More quantification seems to be accessible to the authors given the collected data and would strengthen the claim of the authors. The reviewer proposes 4 major additions:

a. In particular, the manuscript reports showing a typical SEM image of a rosette to show that there is only one flagellated cell. A systematic analysis of more of these images for the number of flagellated cells would provide direct evidence of the authors' claim that these rosettes are only powered by one flagellum (indirectly deduced from the rest of the evidence).

b. Similarly, an analysis of the orientation of the flagellated cell (radial or orthoradial) would help establish the picture of the rosette rotating around the radially oriented axis of the flagellated cells (see Motor reversal minor revision paragraph).

c. From Movie S1, rosettes appear to alternate between two different regimes with almost no movement and then long-range movements that the authors focus on in the second paragraph. A mention and quantification of this feature seems necessary since the effect is quite striking. The data gathered by the authors would allow to characterize these two regimes. Therefore, we encourage the authors to comment and try to explain the phenomenon, and potentially relate it to the dynamics described in Fig. 4.

d. The authors mentioned that they have a way to collect more extensive data about the rosette sizes for future work. Given that the rosette size is a parameter used in their calculations, it would be helpful to report the distribution of the sizes they were able to measure on the presented data.

3. The claim of the no-slip motion does not seem fully supported by the data as presented in this version. A value of $Q=0.5$ is intermediate for a parameter varying from 0 to 1. The conclusion that it is comparable to 1 seems excessive or needs further explanation. Similarly, the measurements of Fig. 3b are largely distributed, beyond the range 0 to 1 and therefore the error bars are large. Nor the change as a function of distance to the surface neither the deviation from the valid

hydrodynamic model are striking and the analysis of this data would benefit from the use of statistical tests to sort out what is significant from what is not.

Minor revisions

Otherwise, I would like to highlight minor modifications that would help clarify some parts of the manuscript, organized by paragraphs:

Introduction

-when reviewing types of motility, it may be useful to indicate, and potentially discuss, the specificities of each of them (e. g. swarming is a surface associated motility)

-detail more why it is important to understand intermediate regimes (motile and non-motile) in general (allocation of resources, emergence...) and in particular (for the studied organism)

-“developmental stages”: explicit more what you are referring to in this context

Sessile-motile coexistence leads to active rosette motilities

-it is unclear how the rosette forms in general and in the authors' experimental conditions. More about this in the main text and in the methods section would be welcomed, in particular addressing if the stalks attach to the surface at first or to each other directly

-refer to figure S6 in the main text when mentioning the average speed of the rosette. Perhaps talking about the “average ballistic speed” would be more clear.

-more details should be provided on how the data with swarmer cells were obtained (in the methods section)

-more details should be provided on how the 2D tracking (Fig. 1b and Movie S1) was performed (in particular in Movie S1, the orange rosette seems to disappear in z and then reappear)

-18 rosettes are displayed in Fig. 1b, but only 9 rosettes tracks are included in the statistics in Fig. 1c. Please provide the rationale behind that in the methods section or include the additional data.

Rosette's rotation in 3D

-histogram of the rosette size would strengthen the estimate given for the calculation

-give the calculated range of values of $L_{\text{rosette}}/L_{\text{solitary}}$ in the main text

-specify that lateral movement is obtained by PIV in the main text

Solid surfaces promotes rosettes' rolling

-when referring to x-y plane, giving a reference with respect to the setup (e. g. bottom plane, surface plane or imaging plane) would clarify the description

-show angle beta on both insets in Fig 3a

-the sentence " We attributed this distribution to the complex rosette shape and the alignments of the flagellar axes, which do not necessarily point toward the rosette center" is confusing. Either give more details or remove. It may be helpful to reason in terms of null hypothesis to clarify the message.

Motor reversals result in reorientation of rolling rosettes

-all along the paragraph, it is assumed that the rosette is rotating around the axis of the flagellum, which is also the conclusion. It would be helpful for clarity to discuss alternative scenarios (e.g. orthoradial flagellum in the periphery creating a torque through leverage) and discuss the evidence that allow to rule out these other hypotheses. In particular, data from the orientation of the flagellum from SEM may be helpful to prove that point.

Discussion

-the mention of "long range" transport is unclear. The reviewer assumed the authors are comparing the range with the range of transport by diffusion only. However, it would also be helpful to compare to other biological scenarios like a single cell reseeding a community elsewhere.

-mention of preliminary data that could benefit being added to this paper, in particular size distribution of rosettes (see major comments)

RESPONSE TO REVIEWER# 1

In their manuscript Zeng and Liu report on the active dispersal and motility of rosette aggregates of *Caulobacter crescentus*. They study the 3D rotational and translational motion of such bacterial aggregates, which consist of both motile and sessile, i.e. non-motile, constituents. In particular, the authors show that the torque generated by a flagellar motor results in a wheel-like rolling motion of aggregates in the vicinity of a solid/liquid interface.

The manuscript, in general, is very well written and the ideas are clearly presented. The experimental approach is described concisely and the results are comprehensively elaborated in the main text, the figures as well as in the supplementary material. In particular, the figures provide clear and visually very appealing representations of experimental data. In addition to the high quality of the manuscript, also the system under investigation is scientifically very interesting. Much has been learned over the past decades on individual motile cells, e.g. spermatozoa, *E. coli* and *B. subtilis*, as well as cellular aggregates and populations of motile cells. However, the motility of multi-cellular entities exhibiting bimodal states of their cellular constituents still remains elusive to date.

In my opinion the work does satisfy the publication criteria of *Communications Biology*. The paper significantly advances the field of motile active matter and it is also of great interest to a general audience. The research has been conducted well and the results are - to the best of my knowledge - entirely novel.

I'd like to raise a few important points that the authors might want to address in their revised manuscript:

We are pleased to hear all the positive comments from the reviewer. In the following, we address the remaining concerns of the reviewer, one at a time.

a) Both individual cells and aggregates exhibit a transition from ballistic to diffusive at around 1 second as displayed in the exemplary MSD curves. What is the physiological meaning of this time scale? Can this be related to a tumble time or a rate at which reorientations occur? The authors should also elaborate more on the propulsion characteristics of individual *C. crescentus* cell (i.e. single flagellum, multiple flagella and bundling like in *E. coli* etc.). The authors speculate that the fact that this transition time is conserved for the aggregates might indicate a common underlying mechanism of propulsion at work. What is known about this mechanism? On page 4 they discuss “run-reverse-flick” motion and some of the above questions. I think this should be introduced earlier in the manuscript since not many readers might be familiar with this type of motility.

We agree that the physiological meaning of this crossover time of the MSD should be explained explicitly. The crossover time for the swarmer cell is associated with the timescale during which the cell is likely to maintain its swimming direction. Beyond that timescale, the cell is likely to turn randomly into a new direction, here, through a two-step run-reverse-flick process (due to the single-flagellum setting of *C. crescentus*). The exact physiological meaning of this crossover time to a rosette is less clear due to its underexplored kinematics, which we tried to illuminate in this study. Overall, this timescale should also reflect the reorientation rate of rosettes. We also agree that the reorientation mechanisms of a single swarmer cell should be introduced earlier. We have now discussed the physiological meaning of the crossover time and introduced the run-reverse-flick mechanisms earlier in our manuscript:

(In Results, **Sessile-motile coexistence leads to active rosette motilities**, paragraph 4, line 7) “In self-propelled particles, such a crossover lag time between ballistic and diffusive regimes is typically associated with the duration of self-propulsion, ... Similarly, propulsion with a single flagellar motor likely underlies the motility of rosettes despite the substantial size difference.”

(In Introduction, paragraph 3, line 7) “A motile cell is propelled by the rotation of a single helical flagellum, ... These forward-reverse-flick switches lead to a random-walk mechanism of *C. crescentus* distinct from that of many peritrichous bacteria (e.g., *Escherichia coli*), which rely on run and tumble to swim and reorient [25].”

b) Rotational degrees of freedom: The authors neglect any boundary effects on the rotational motion and, in particular, the torque calculations. What about adhesion and friction of sessile cells at the glass surface? Are the aggregates hovering above the the glass slide or is there any evidence of direct contact at the solid/liquid interface? Provided there is friction, one might need to consider more than just one motile cell being involved in rotation la motion.

We did neglect the boundary effects in this scaling analysis of rotation in the first attempt. Indeed, the purpose of this analysis is to show that the torque from a single flagellar motor can cause a significant rotation of rosettes, comparable to our experimental observation. We did not rely on this analysis to argue that all rotational movements of rosettes in this study were driven by a single motor.

Also, we have considered the wall effects (through hydrodynamic interactions) in our scaling analysis. For a rotation axis perpendicular to the solid surface, the additional torque due to the presence of the solid surface is almost

negligible, as concluded by both a boundary element simulation (see *SI*) or a lubrication theory (E. Lauga, et al., *Biophys J.* **190**, 400–412, 2006). For a rotation axis parallel to the solid surface, the additional torque due to the wall surface diverges as the gap between a smooth sphere and the wall vanishes. However, for a finite gap between the rosette and the wall d , the increase of torque due to the surface is still a fraction of the torque in the free-space case, e.g., the torque increased only by 40% than the free-space case for a gap-to-radius ratio $d/R = 0.1$. Further, through a hydrodynamic simulation of the rosette kinematics (included in the revised *SI*), we show that the latter rotation configuration (with a rotation axis parallel to a nearby surface and thus relatively larger torque) is unstable. The additional drag due to the surface also gives rise to a torque that tends to orient the rotation axis perpendicular to the wall (see new supplementary movie S6). For the above reasoning, we believe that our scaling analysis is still decent without hydrodynamic consideration of the wall effects. It should be noted that we still can not rule out the surface effects due to other interactions, such as adhesion, which, however, cannot be fully addressed in the scope of this work. We have now included the following statement in our main text:

(In Results, **Rosettes’ rotation in 3D**, paragraph 2, line 12) “It is worth noting that we considered only the free-space situation in the above scaling analysis. However, including the hydrodynamic effects of a nearby wall will not alter our conclusion, as verified by a hydrodynamic simulation (see *SI*).”

c) Can the size of an aggregate be adjusted experimentally - e.g. via culture media and/or cultivation procedures? In other words, which parameters determine the number of cellular constituents forming one aggregate?

Yes, it has been established that the size range of rosettes depends on the culture condition, such as the initial concentration of viable cells (J. S. Poindexter, *Bacteriol. Rev.* **28**:231-95, 1964). We speculate that this dependency also determines the distribution of the rosette sizes ($R = 2\text{--}6\ \mu\text{m}$) in this study, included now in the revised *SI*. We have also included more details of the culture method in the Methods to specify the culture condition that our rosette sizes are subjected to.

In addition, we speculate that the common mass of adhesive holdfast at the rosette center (required for rosette formation) also potentially limits the maximum number of member cells for a rosette. As the reviewer will find in our response to the next comment, we argue that this finite number of member cells also contributes to a mechanism that a single-flagellum propulsion dominates in this study.

d) Motility mutants: Is there a way to test the hypothesis that an aggregate is powered by precisely one flagellar motor? Are motility mutants of *C. crescentus* with distinct motility characteristics available, which could then be used in analogous experiments?

We did not intend to argue that the rosette is precisely driven by one flagellar motor. Both the MSD (Fig. 1c) and the motor duration (Fig. 4c) indicate a single functioning motor from the statistical point of view. However, we cannot provide direct visual proof to support this argument. It is also a great suggestion from the reviewer to use motility mutants of *C. crescentus* for hypothesis tests. However, the only motility mutant that we are aware of is a mutant that can only swim in the forward direction (due to a fixed CW motor) (G. L. Li and J. X. Tang, *Phys. Rev. Lett.* **103**: 078101, 2009). While we are still uncertain how this type of mutant can be useful in hypothesis testing, we have sought a probability-based understanding of this single-motor mechanism, making use of the origin of powering sources in rosettes. Considering each stalked cell can possibly acquire a flagellar motor (equipped on the daughter cell) for a limited period of time during its division, we can define a probability for finding a flagellated stalked cell, which is given by the flagellated duration over the duration of each division cycle. By estimating both timescales, we show that such a probability is on the order of 1%. Using this probability, we can thus estimate the probability of finding a given number of flagellar motors within a rosette. Given the finite size of the rosette in this study (with the number of member cells typically less than 25), the probability for finding only one motor within a motile rosette is much higher than other scenarios. We have now elaborated this statistical model and other possible scenarios in a new section **Potential mechanisms (or biases) for single-motor powering of rosette**.

e) Collective effects: Have the authors performed experiments at high aggregate densities? How do two (rotating) aggregates interact and do they form coupled states of motility (see, e.g., *Volvox* colonies in the vicinity of a glass surface)?

A systematic study of the collective motility of rosettes is on our to do list. So far, we have focused on understanding first the motility of individual rosettes as we believe it overambitious to include additionally the interaction of rosettes in one manuscript. We have, however, mentioned this interesting direction of extending of our current scope of work in the Discussion section:

(In Discussion, paragraph 3, line 7) “In addition to these wall effects, the rosette-rosette interaction will also be examined in our future work to understand any collective motility.”

For the reasons outlined above, I strongly support publication of the manuscript in *Communications Biology*, after my questions and remaining points of criticism have been adequately addressed.

We hope that the above responses have addressed most of the reviewer’s concerns and thank again the reviewer’s strong support.

RESPONSE TO REVIEWER# 2

Zheng and Liu are reporting an interesting observation that the motion of *Caulobacter* rosettes is powered by a flagellated cell. One caveat of the study appears to be the assumption that motile rosettes have only one flagellated cell. The data that supports this assumption is not sufficient (only one EM image), yet the authors find that the duration of flagellar motor activity between two consecutive switches exhibits similar frequency distributions for rosettes and solitary cells. This indicates that either there is a mechanism that allows only one flagellated cell per rosette or somehow the authors have developed a bias in their data collection towards rosettes that only have one flagellated cell. My specific comments are below:

We are pleased to find the reviewer find our observation interesting. We should note that we did not intend to argue in this work that the rosette is precisely driven by one flagellar motor. Indeed, our analyses of the MSD (Fig. 1c) and the motor duration (Fig. 4c) unanimously indicate a favorable single-motor mode of rosettes, from a statistical point of view.

We have now devised a statistical model to understand such a single-motor mechanism, making use of the origin of powering sources in rosettes. Considering each stalked cell can possibly acquire a flagellar motor (equipped on the daughter cell) for a limited period of time during its division, we can define a probability for finding a flagellated stalked cell, which is given by the flagellated duration over the duration of each division cycle. By estimating both timescales, we show that such a probability is on the order of 1%. Using this probability, we can thus estimate the probability of finding a given number of flagellar motors within a rosette. Given the finite size of the rosette in this study (with the number of member cells typically less than 25), the probability for finding only one motor within a motile rosette is much higher than other scenarios. We have now elaborated this statistical model and other possible scenarios in a new section **Potential mechanisms (or biases) for single-motor powering of rosette**.

1. On an average, how many motile cells are found in a rosette? What is the probability of observing only one motile cell in a rosette? Since the authors presumably have multiple EM images collected without a selection bias, they can provide the statistics.

While the SEM provides us visual proofs that a flagellated predivisional cells can indeed exist within a rosette, we found it unreliable in counting the number of flagellated cells for the following reasons. First, all SEM images are two dimensional. Any flagellum underneath the rosette is invisible. Second, the SEM sample preparation process was supposedly detrimental to the thin flagella, as has now been elaborated in the Methods. The multiple washes and freeze-drying procedures may damage flagella, as suggested from all our SEM images of rosettes: all flagella observed were incomplete short filaments. Such broken flagella are also convoluted with premature flagella, which makes the counting even more challenging.

We have now included more typical SEM images of rosettes in *SI* to better support the above arguments. We have also included the detailed SEM procedure to illustrate that it cannot be reliably used for flagellar counting. Also, as shown in our response above, we have now included a new section to examine the number of functioning motors of each rosette from the statistical point of view. From the flagellated probability ($P_f \sim 1\%$ as discussed previously), we have estimated the number of functioning motors based on the typical number of member cells ($N_s \lesssim 25$) of a rosette, which should be $N_s P_f \lesssim 1$. This reasoning has also been included in our revised manuscript (with its elaboration in *SI*):

(In Results, **Potential mechanisms (or biases) for single-motor powering of rose**, paragraph 2) “The size of the rosettes may also limit the number of available flagella given the relatively small fraction of predivisional stage in each cell’s life cycle. Based on an ideal statistical model and a known division timescale, $\Delta t_d \sim 10^2$ min [30, 40], we found that, for rosettes in this study ($R = 4.0 \pm 1.2$ μm ; mean \pm S.D.; $N_s \lesssim 25$), the probability of having more than one active flagellum is as low as 3% (see *SI*).”

2. How does positioning of multiple motile cells impact motion of the rosette? Are there rosettes where no motility is observed despite the presence of motile cells? Does positioning of motile cells at opposite ends of the rosette cancel out/decrease net motility or rolling of the rosette?

We expect that such a multiple-flagellum scenario should exist in our study since the single-motor powering is only statistically favorable. It is an interesting question whether the rosette motility can be cancelled when the only two flagella are aligned in the opposite directions. We believe that there can only be a temporary cancellation that lasts for a very short duration ($\sim 1\text{s}$) due to the fact that each motor of *C. crescentus* randomly switches its running direction (CW and CCW). As a consequence, the rosette activity is expected to frequently switch between a cancellation motility (same motor directions) and a double motility (opposite motor directions), which are not noticeable to us in

this study. We have now included a discussion on multiple-flagellum powering in our manuscript:

(In Results, **Potential mechanisms (or biases) for single-motor powering of rose**, paragraph 1, line 2)“ This single-motor priority may potentially be limited by the scope of this work, covering only motile rosettes and their movements near the bottom surfaces. . . . Such fluctuations in the number and location of effective motors conflicts with the persistent rotational movements observed in this study. ”

3. The authors provide a calculation suggesting that a single motor can power a rosette’s rotation but they don’t tell us the prevalence of this scenario. One way to test it might be to fluorescently label the flagellum and count the number of flagellated cells in motile rosettes.

We have considered this option of labeling the flagellum. However, the current labeling approach appears to have a low success rate on *C. crescentus* without impacting its motility (P. P. Lele, *et al.*, Nat. Phys. **12**, 175–178, 2016). However, beyond the previous order-of-magnitude calculation, we have now included in *SI* a more complete hydrodynamic model (including both a spherical rosette and a helical flagellum) to demonstrate the capacity of a single flagellum in rosette propulsion. We have also included two new supplementary movies to show simulated results with two extreme cases of flagellar alignments (radial and orthoradial). While the simulations resemble some experimental observations, we regard their detailed comparisons suitable for a separate publication.

4. Could the authors provide videos of rosettes near and far from a solid surface and compare their rolling? This might make their statement that ‘solid surfaces promote rosettes rolling’ stronger.

We have now included a new supplementary movie (movie S4) to exemplify the rosette movements far from the surface.

RESPONSE TO REVIEWER# 3

The manuscript “Self-propelling and rolling of a sessile-motile bacterial aggregate” describes self-propulsive motion of aggregates of active and passive cells of *C. crescentus* which forms a rosette like morphology. Specifically, the authors have pointed out an interesting observation that the motility of the aggregate is basically powered by a single flagella motor. The authors have compared the statistics of a solitary cell with a sessile-motile aggregate and found that the rosette motion follows intermediate values of diffusivities and rotation. The effect of solid boundary and motor reversal effects are also demonstrated. How a passive cluster of cells can show long range active motion by the presence of motile units is the key result of the present work. However, there are few points which need to be addressed before the manuscript can be published.

We thank the reviewer for the useful comments, which are addressed in the following, one at a time.

1. The underlying dynamical process and mechanism of the self-propulsive behavior needs to be explored in detail. How a single motor can bear such a load of sessile aggregate to rotate and roll is not very clear. The author should provide a proper mechanistic explanation of the origin of such motion to move a big unit of aggregated cell.

We should note that although a rosette rotates and rolls, it does so at a much slower speed as compared to an individual swarmer cell. As long as the flagellum on a rosette is untethered, the flagellar motor is expected to operate at a rotation speed and torque similar to that of a swarmer cell. The torque-free condition for a microscale swimmer (including all cell bodies and the flagellum) requires the rosette to rotate in a direction apposing that of the motor. Given the same motor torque, the apparent larger payload for a rosette gives rise to its slower rotational speed. We have now included the above explanation in our scaling analysis:

(In Results, **Rosettes’ rotation in 3D**, paragraph 2, line 12) “It is worth noting that we considered only the free-space situation in the above scaling analysis. However, including the hydrodynamic effects of a nearby wall will not alter our conclusion, as verified by a hydrodynamic simulation (see *SI*).”

2. How significant is the roll of hydrodynamics interaction with the surrounding medium for the active dispersal?

We believe that the hydrodynamic interaction with the surrounding medium is the key for the torque and thrust generation of an active rosettes, especially when rosettes are far from a solid surface. We have now included a hydrodynamic model of rosette with the presence of solid surfaces (in *SI*, **Hydrodynamic simulation of rosette kinematics**, supplementary movies 6 and 7) to demonstrate the roles of hydrodynamic interaction in rosettes’ dispersal.

3. For a dense system of cells, is there any suppression of the self-propulsive rotational motion? Or, bigger aggregates of sessile and motile cells may emerge in dense system. Specifically, with the variation of cell number density how the morphology and spatiotemporal dynamics changes?

We are yet to observe any suppression of the rotational movements of a rosette for dense member cells. This may be partially due to the narrow distribution of rosette sizes in this study, which presumably depends on our culture protocol. It has been established that rosette size depends on the initial concentration of viable cells (J. S. Poindexter, *Bacteriol. Rev.* **28**:231-95, 1964). We thus expect that larger rosettes may emerge in cultures with higher initial concentrations. The potential dependence of a rosette’s morphology and spatiotemporal dynamics on cell number densities (as brought up by the reviewer) is an interesting topic by itself, which we may explore in future works. However, we have included the distribution of rosette sizes (in *SI*, Fig. S11) and our detailed culture method to show the standing of the current work:

(In Methods, **Swarmer-cell and rosette preparation**, paragraph 2, line 1) “Two methods were used to obtain rosettes in this study. . . ., which were presumably formed in the culture mixed with stalked cells and predivisional cells.

Contribution

The authors study the motility of an assembly of motile and sessile cells, the rosettes of *C. crescentus*. They use SEM and phase contrast microscopy to visualize the rosettes and analyze their motion. Using visualization, a 3D reconstruction of the motion, torque estimations and trajectory analysis, they deduce that the rosettes are powered by a single flagellated cell that creates a tumbling motion when the rosettes are close to the surface.

Summary

Overall, the manuscript is well written and the data is supporting the claims. The experimental methods are thoroughly executed and novel, in particular the 3D tracking of the rosette motion. Three major (but nonetheless easily accessible to the authors) points would help perfect this manuscript to make it worthy of a publication in *Communications Biology*:

We are pleased to hear that the reviewer found our manuscript “well written” and our experiments “thoroughly executed and novel.” We also thank the reviewer for the insightful and instructive comments, which we address in the following, one at a time.

1. More context about the *C. crescentus* rosette should be provided necessary to better frame the significance of the study, its contribution to the field. In particular, it is currently unclear how much this system has been described and studied in the past (not clear by looking at the references). More on the emergence (e. g. conditions of emergence, prevalence in natural settings) of these rosettes would also be helpful. The reviewer therefore recommends to remodel the third paragraph of the introduction to better put into context the findings of this biophysical study for the biology oriented readership of the journal.

To our best knowledge, *C. crescentus* rosette has always been treated as a on-motile body in earlier studies. Although noted by works on cell adhesion, the adaptive significance of *C. crescentus* rosettes is underexplored. We thus believe that our work is the first attempt in understanding the significance of rosettes from a dispersal perspective. Though lacking field studies, *C. crescentus* rosette has been regularly reproduced under the laboratory conditions (J. S. Poindexter, *Bacteriol. Rev.* **28**:231-95, 1964), which suggests the prevalence of *C. crescentus* rosette or its similar forms in nature, especially considering that *Caulobacter* species is widespread in soil and aqueous environments (including tap water). We have now elaborated more of such contexts by remodeling the third paragraph and as well as introducing an additional (the fourth) paragraph:

(In Introduction, paragraph 3, line 2) “*Caulobacter* bacteria are widespread in soil, aqueous environments, within organisms, as well as in clinical systems (e.g., tap water and millipede body ... However, its biological significance and motility of rosettes are unclear.”

2. More quantification seems to be accessible to the authors given the collected data and would strengthen the claim of the authors. The reviewer proposes 4 major additions:

a. In particular, the manuscript reports showing a typical SEM image of a rosette to show that there is only one flagellated cell. A systematic analysis of more of these images for the number of flagellated cells would provide direct evidence of the authors’ claim that these rosettes are only powered by one flagellum (indirectly deduced from the rest of the evidence).

We primarily used the SEM image to demonstrate the possibility of containing motile cells within a rosette aggregate. However, we found the SEM unreliable for quantifying the number of flagella in a living rosette for the following reasons. In order to prepare an SEM specimen, the sample has been freeze-dried and coated by gold for reflecting electrons. The process is expected to be detrimental and very likely to reduce the observed number of flagella. In addition, the SEM images are essentially two dimensional projections from a top view, which does not allow us to view any flagella underneath the rosettes. This two-dimensional limitation of SEM thus also prevents us from using it for flagellar counting. We have now included more SEM images (in *SI*, Fig. S12) to support the above argument. We have now also included the detailed process of SEM (in Methods, **Scanning electron microscopy**) to show that the SEM alone cannot be used to verify the number of functioning flagella of rosettes.

While we cannot directly visualize the number of functioning flagella (e.g., through SEM), we have performed another independent analysis of the probability for the rosette to contain more than a single flagellum. Here, we make use of a probability of finding a member cell contain a functioning flagellum through division and the typical number of member cells found in this study (see more details in our responses to “major addition c”). We believe that such an analysis can provide an explanation for why single-motor powering is preferable for rosettes in this study.

b. Similarly, an analysis of the orientation of the flagellated cell (radial or othoradial) would help establish the picture of the rosette rotating around the radially oriented axis of the flagellated cells (see Motor reversal minor revision paragraph).

For the same reason shown in our response to the reviewer’s “major addition a”, we cannot rely on SEM for characterizing the alignment of flagella. Labeling the flagella is potentially an alternative for direct visualization. However, such a technique is surprisingly unstable for *C. crescentus*, which causes the cell to become immotile (P. P. Lele, *et al.*, Nat. Phys. **12**, 175–178, 2016) and is likely to yield a biased result. However, we have performed new simulations (through hydrodynamic modeling) to show the roles of flagellar alignment in rosette movements, which is elaborated in our response to the reviewer’s “Motor reversal minor revision” in the following.

c. From Movie S1, rosettes appear to alternate between two different regimes with almost no movement and then long-range movements that the authors focus on in the second paragraph. A mention and quantification of this feature seems necessary since the effect is quite striking. The data gathered by the authors would allow to characterize these two regimes. Therefore, we encourage the authors to comment and try to explain the phenomenon, and potentially relate it to the dynamics described in Fig. 4.

We thank the reviewer for bringing up this important point that we left unaddressed in the previous version of our manuscript. We speculated that the alternation between two types of rosette activeness was associated with the dynamics of *C. crescentus* division: the rosette acquires active motility when it exploits any functioning motors and flagella from predivisional member cells; it diffuses passively when motors and flagella are either not yet available among all member cells (e.g., after detachment of all new born cells) or not functioning. Assuming that each of such consecutive movements is associated with an individual set of functioning motors, we can estimate the probability for finding a given number of motors within a rosette. More specifically, the duration of each consecutive active movement provides a measurement of the maximum duration that each functioning flagellum can be employed by the rosette, which is on the order of $\Delta t_f \sim 1$ min. Given that each stalked cell takes about $\Delta t_d \sim 10^2$ min (S. T. Degnen and A. Newton, J. Mol. Biol., **64**: 671-680, 1972; a Iyer-Biswas *at el.*, PNAS, **111**:15912-15917, 2014) for dividing into a new born cell, the probability of containing a functioning flagellum (during division) for each stalked member can be well estimated by $P_f = \Delta t_f / \Delta t_d \lesssim 0.01$. Assuming unsynchronized division of each member, we can now investigate the probability for the whole rosette (containing N_s stalked member cells) to have n functioning flagella, as formulated by $P(n, N_s) = \binom{N_s}{n} P_f^n (1 - P_f)^{N_s - n}$. For the typical rosette sizes in this study ($N_s \sim 15 - 25$), we found that the rosette is likely to contain none or one flagellum, with the probability of the latter case [$P(1, N)$] almost one order of magnitude greater than the probability of all other scenarios (e.g., $n > 1$). This calculation thus further addresses our speculation that most rosettes in this study are powered by a single flagellum.

We have now included these new data (in SI, Fig. S13), analyses (in Results, **Potential mechanisms (or biases) for single-motor powering of rosettes**), and calculation (in SI, **Statistical model for the number of functioning motors in a rosette**).

We have also mentioned these two motility regimes and emphasized that the rest of this work focuses on the motile regime of the rosette in the main text:

(In Results, **Sessile-motile coexistence leads to active rosette motilities**, paragraph 3, line 2) “Over a sufficiently long period of time ($\gtrsim 10$ min), the trajectories of rosettes were often interrupted by noticeable pauses, during which rosettes appeared almost immotile (speed $u \lesssim 0.5 \mu\text{m/s}$).” This coexistence of both motile and immotile regimes is presumably resulted from the dynamics of its powering source during cell division. In the following, we only focused on this motile regime.”

Any potential reorientation mechanisms during the non-motile regime (e.g., due to diffusion) are thus out of the scope of this study (Fig. 4).

d. The authors mentioned that they have a way to collect more extensive data about the rosette sizes for future work. Given that the rosette size is a parameter used in their calculations, it would be helpful to report the distribution of the sizes they were able to measure on the presented data.

We have now included a distribution of rosette sizes (in SI, Fig. S11) and used the range of rosette sizes in our scaling analysis of the torque on a rosette:

(in Results, **Rosettes’ rotation in 3D**, paragraph 2, line 5) “Given rosettes in this study ($R = 2 - 6 \mu\text{m}$, $e = 0$; see SI) ... the ratio between the torque on the rosette and that on a single cell $L_{\text{rosette}}/L_{\text{swarmer}} = 0.2 - 10$.”

3. The claim of the no-slip motion does not seem fully supported by the data as presented in this version. A value of $Q=0.5$ is intermediate for a parameter varying from 0 to 1. The conclusion that it is comparable to 1 seems excessive

or needs further explanation. Similarly, the measurements of Fig. 3b are largely distributed, beyond the range 0 to 1 and therefore the error bars are large. Nor the change as a function of distance to the surface neither the deviation from the valid hydrodynamic model are striking and the analysis of this data would benefit from the use of statistical tests to sort out what is significant from what is not.

We agree with the reviewer that claiming the no-slip motion was not fully justified by our data. We have now performed more statistics on these data. Through both linear and non-linear regressions, we show that the dependence of the $u - \omega$ ratio on the gap size is significant (using the 95% confidence bounds). This dependency (with the 95% confidence bounds) does not have any overlap with the prediction from the hydrodynamic model on a smooth sphere, suggesting that this idealized hydrodynamic model alone cannot fully explain the measured $u - \omega$ ratios near the wall. We have also performed other significance analyses, which all lead us to similar conclusions. We have now included the result of the non-linear regression in Fig. 3b.

Its detailed method and other significance analyses have now been included (in *SI*, **Statistical tests of the slipping ratio** Q and Fig. S10). We have also removed the phrases “which is comparable to the no-slip case ($Q = 1$)” as our current data appear not to support this argument.

Minor revisions

Otherwise, I would like to highlight minor modifications that would help clarify some parts of the manuscript, organized by paragraphs:

Introduction

-when reviewing types of motility, it may be useful to indicate, and potentially discuss, the specificities of each of them (e. g. swarming is a surface associated motility)

We have now included more specificities of each types of motility:

(In Introduction, line 2) “In active dispersal, individual cells are equipped with motile organelles, such as flagella or pili, which enable swimming (in liquids) or gliding motilities (near a surface), respectively. In dense populations of motile cells, cell-to-cell interactions lead to collective dispersal, such as three-dimensional vortical flows in bulk fluids and quasi two-dimensional swarming above a semisolid surface, e.g., an agar plate. Also, collective motility can be achieved through aggregation, maintained through intercellular coalescence, as larger dispersal units.”

-detail more why it is it important to understand intermediate regimes (motile and non-motile) in general (allocation of resources, emergence...) and in particular (for the studied organism)

We have now included the reasoning for studying the intermediate dispersal regime and the specific bacterial species:

(In Introduction, paragraph 2, line 6) “ In this intermediate regime, the motile compartment of the dispersal unit is in principle capable of carrying the entire aggregate and allocating resources.

To investigate this intermediate mode of dispersal, we explored the motility in rosette aggregates of an aquatic bacterium: *Caulobacter crescentus*. *Caulobacter* bacteria are widespread in soil, aqueous environments, within organisms, as well as in clinical systems (e.g., tap water and millipede body). Gaining a full understanding of *Caulobacter* motility thus impacts many ecological and medical applications. ”

-“developmental stages”: explicit more what you are referring to in this context

We have now replaced the term by “possessions of motile organelles due to asynchronous development in life cycles” to make it more explicit.

Sessile-motile coexistence leads to active rosette motilities

-it is unclear how the rosette forms in general and in the authors’ experimental conditions. More about this in the main text and in the methods section would be welcomed, in particular addressing if the stalks attach to the surface at first or to each other directly

Throughout our light-microscopy observation, we did not notice any ongoing rosette formation. This visual absence thus suggested that the rosettes were formed in the agitated growth media during the culturing process. Though not visualized directly, it has been further speculated by Poindexter (J. S. Poindexter, *Bacteriol. Rev.* **28**:231-95, 1964) that these rosettes are formed by random collisions of suspended stalked cells at similar growth stages, in the absence of surfaces. Meanwhile, rosettes have been typically found in agitated culture liquid, with their sizes dependent on the initial concentration of viable cells (J. S. Poindexter, *Bacteriol. Rev.* **28**:231-95, 1964). This dependency on culture condition is also consistent with the narrow distribution of rosette sizes that we observed in our study. We have now included in Methods the detailed conditions for rosette culture:

(In Methods, **Swarmer-cell and rosette preparation**, paragraph 2, line 1) “Two methods were used to obtain rosettes in this study. . . ., which were presumably formed in the culture mixed with stalked cells and predivisional cells. ”

We have also included the above understanding of rosette formation in the main text:

(In Introduction, paragraph 4, line 4) “Though lacking visual evidences, it has been speculated that the rosettes are formed by random collisions of stalked cells at similar growth stages [20], which then attach to a common core through holdfast [26, 28]. ”

-refer to figure S6 in the mean text when mentioning the average speed of the rosette. Perhaps talking about the “average ballistic speed” would be more clear.

We have followed the reviewer’s suggestion and revised the “average speed” as the “average ballistic speed.”

-more details should be provided on how the data with swarmer cells were obtained (in the methods section)

We have now specified how the data with swarmer cells were obtained in Methods.

(In Methods, **Swarmer-cell and rosette imaging**, line 5) “For *C. crescentus* swarmers, the microscope stages were programmed to move along with an individual cell in 3D to maximize the tracking duration [33]. The position of each cell was thus recorded from the position of the microscope stage.”

-more details should be provided on how the 2D tracking (Fig. 1b and Movie S1) was performed (in particular in Movie S1, the orange rosette seems to disappear in z and then reappear)

We have now included more detailed tracking of rosettes in Methods:

(In Methods, **Swarmer-cell and rosette imaging**, line 9) “Rosette images were subjected to a zeroth order Bessel function and a Gaussian filter such that each rosette appears as a simply connected object (same as we used for axial position reconstruction, see *SI*). The 2D position of a rosette in each frame was thus determined by the centers of the above filtered object.”

The orange track captured by the reviewer indeed belongs to two different rosettes, which happened to be coded by the same color (due to the limited number of sharp colors that we applied). We have modified the color of one of the two rosettes to reflect this difference. We also labeled each rosette by a unique number in the video (revised Movie S1) to make each trajectory distinguishable.

-18 rosettes are displayed in Fig. 1b, but only 9 rosettes tracks are included in the statistics in Fig. 1c. Please provide the rationale behind that in the methods section or include the additional data.

Only those rosettes that stayed active within the view for over a minute were used for MSD calculation (Fig. 1c). Not all 18 rosettes in Fig. 1b satisfy these criteria, which leads to discrepancy in number of tracks between Fig. 1b and Fig. 1c. We have now provided this rationale in Methods.

(In Methods, **Rosette dispersal**, line 4) “To avoid the potential bias from short rosette tracks, rosettes that stayed active within the view for over a minute were used for MSD calculation (Fig. 3c). Not all 18 rosettes in Fig. 3b satisfy this criterion, which leads to discrepancy in numbers of tracks between Fig. 3b and Fig. 3c. ”

Rosette’s rotation in 3D

-histogram of the rosette size would strengthen the estimate given for the calculation

We have now provided the histogram of the rosettes’ size (in *SI*, Fig. S11).

-give the calculated range of values of $L_{\text{rosette}}/L_{\text{solitary}}$ in the main text

We computed the ratio between the torque on the rosette and that on a single cell $L_{\text{rosette}}/L_{\text{swarmer}}$ based on the scaling analysis, which is within the range of 0.2 – 10. It is worth noting that the large variation of the ratio (almost by two orders of magnitude) is due to the ignorance of the size dependency in the rotational speed of the rosette, i.e., a larger rosette should be associated with a lower rotation rate. The following has been included in the main text:

(in Results, **Rosettes’ rotation in 3D**, paragraph 2, line 5) “Given rosettes in this study ($R = 2 - 6 \mu\text{m}$, $e = 0$; see *SI*) . . . the ratio between the torque on the rosette and that on a single cell $L_{\text{rosette}}/L_{\text{swarmer}} = 0.2 - 10$.”

-specify that lateral movement is obtained by PIV in the main text

We have specified the lateral movements by including “we obtained the lateral movements of rosettes near the image plane through particle image velocimetry (PIV)” in the main text.

Solid surfaces promotes rosettes’ rolling

-when referring to x-y plane, giving a reference with respect to the setup (e. g. bottom plane, surface plane or imaging plane) would clarify the description

We have now referred the x - y plane to the “bottom surface.”

-show angle beta on both insets in Fig 3a

We have show shown the angle beta for both insets.

-the sentence “ We attributed this distribution to the complex rosette shape and the alignments of the flagellar axes, which do not necessarily point toward the rosette center“ is confusing. Either give more details or remove. It may be helpful to reason in terms of null hypothesis to clarify the message.

We meant that “the complex rosette shape and the alignment of the flagellar axes” give rise to a rich variety of angles between the rotation and translation axes. Moreover, even for fixed angle between these two vectors, the projection of this angle to the x - y plane is arbitrary for a rosette swimming in the bulk, subjected to the orientation of the plane formed by these two vectors. We have now included the above argument in our text:

(In Results, **Solid surfaces promote rosettes’ rolling**, line 6) “Such a complex configuration give rise to a rich variety of angles between the rotation and translation axes. Moreover, even for fixed angle between these two vectors, the projection of this angle to the x - y plane is arbitrary for a rosette swimming in the bulk, subjected to the orientation of the plane formed by these two vectors.”

Motor reversals result in reorientation of rolling rosettes

-all along the paragraph, it is assumed that the rosette is rotating around the axis of the flagellum, which is also the conclusion. It would be helpful for clarity to discuss alternative scenarios (e.g. orthoradial flagellum in the periphery creating a torque through leverage) and discuss the evidence that allow to rule out these other hypotheses. In particular, data from the orientation of the flagellum from SEM may be helpful to prove that point.

In this paragraph, we only consider that the direction of rosette circulation is opposite to the rotation direction of the flagellar motor, based on the surface effect due to hydrodynamic interactions. The work that we previously referred to (E. Lauga *et al.*, *Biophys. J.*, **90**: 400–412, 2006) did focus on the circumstance with the flagellar axis along the radial direction of a spherical payload and the swimmer swimming parallel to the solid surface. Based on these assumptions, it is much easier to show the direction of the additional torque due to the proximity of the solid surface, which gives rise to the circular movement of cell opposite to the rotation direction of the flagellar motor.

To show whether this direction of rosette circulation can be used to signify the direction of flagellar motor (for arbitrary flagellar axis and swimming directions), we have employed a similar hydrodynamic model except allowing for arbitrary flagellar alignments and swimming directions. We found that the direction of circulation as an indicator for motor direction is indeed universal for rosettes moving near the solid surface. However, the detailed mechanisms are much more complicated than what can be shown by a free-body diagram, due to the time-dependent flagellar axis relative to the solid surface.

We have now included this numerical modeling (in *SI*, **Hydrodynamic simulation of rosette kinematics**), and demonstrated the aforementioned relation of the motor direction and the circulation direction by two supplementary movies (Movie S6 and S7), showing the two extreme cases with the flagellar axis along the radial and orthoradial directions, respectively. For the former (radial) case, we also show that the situation with flagellar axis parallel to the surface is unstable, which becomes perpendicular to the surface due to the hydrodynamic interactions. While we found that the numerical simulation resembles similar rosette kinematics that we observed in this study, we believe that it deserves a separate publication (that focuses on detailed rosette propelling mechanisms) for more quantitative comparison between the experimental data and the model predictions. We have also included the above clarification in the main text:

(In Results, **Motor reversals result in reorientations of rolling rosettes**, line 5) “We should note that here we assumed that the flagellar axis is along the radial direction of the rosette (see *SI* for cases with more general flagellar alignments, e.g., along the orthoradial direction).”

In addition, we believe our previous conclusive message is not delivered clearly, which appears as a loop hole in our reasoning. We meant that the unanimous sharp turning of the rosette was an indicator that it is not propelled parallel to the direction of the axis of flagellar motor. Otherwise, the rosette reorientation will be subject to a hook buckling during a CCW-to-CW motor switch, resulting in a flick in rosettes' reorientation similar to the solitary cell case. We have now included these arguments in our text for clarifications:

(In Results, **Motor reversals result in reorientations of rolling rosettes**, paragraph 2, line 8) “In contrast, the unanimous sharp turning of the rosette is an indicator that it is not propelled parallel to the direction of the axis of flagellar motor. Otherwise, the rosette reorientation will be subject to a hook buckling during a CCW-to-CW motor switch, resulting in a flick in rosettes' reorientation similar to the solitary cell case. We thus speculated that rosette rolling was along the direction of the flagellar torque, with its direction of translation perpendicular to its motor axis (Fig. 4a, inset). ”

Discussion

-the mention of “long range” transport is unclear. The reviewer assumed the authors are comparing the range with the range of transport by diffusion only. However, it would also be helpful to compare to other biological scenarios like a single cell reseeding a community elsewhere.

We agree with the reviewer that the definition of “long-range” is unclear here. As the reviewer mentioned, we meant to refer this long range to the quantitative comparison of the range of rosette dispersal with that through diffusion. We would also like to compare the range of rosette transport to other biological scenarios. Limited by the tracking time of individual rosettes, we believe that such a range in this study ($\sim 100\mu m$) is much shorter than that can be achieved by a single motile cell for reseeding purposes. However, since the rosette harvests the power source of predivisional members, we believe the rosette may exploit many generations of daughter cells for its dispersal, as long as its member cells continuously divide. This “long-range” transport cannot be captured in this study and require a separate experiment for much longer rosette tracking. To make our description more rigorous, we have removed the word “long-range” from the text.

-mention of preliminary data that could benefit being added to this paper, in particular size distribution of rosettes (see major comments)

We have now included the size distribution of rosettes that are gathered under our experimental conditions (Fig. S11).

FIG. 1. Updated Fig. 3. Angle β is shown in both insets of **a**. A nonlinear regression of the data (sampled every second) with 95% confidence bounds is shown in both **b** and its inset.

FIG. 2. New Fig. S9. Schematic of the hydrodynamic model of an actively propelling rosette near a wall that lies in the x - y plane of a lab frame of reference (x, y, z) . The rosette geometry is estimated as a smooth sphere (to represent all cell bodies aggregated within the rosette) of radius R and a finite helical filament [?] of the same dimension of the flagellum [?] possessed by a swarmer cell. The spherical cell aggregate and the helical flagellum are centered at O_c and O_f , respectively and intersect at O' . Their combined dynamics are examined by a resistive-force-type approach [?], with the forces \mathbf{F} and torques $\mathbf{\Gamma}$ on both objects (sphere and helix) solved independently and satisfying the force- and moment-balance conditions. In this model, a body-fixed frame of reference (x', y', z') is conveniently defined by aligning the x' axis along the helical axis of the flagellum and confining the separation vector between the aggregate and the flagellum ($O_c O_f$) in the x' - z' plane. The deviation of the flagellar axis from the radial direction is accounted for by introducing an alignment angle θ_{cf} , the angle between the vector $O' O_c$ and the x' axis. The angular velocity of the helix $\boldsymbol{\omega}_f$ relative to the sphere $\boldsymbol{\omega}$ is prescribed in the body-fixed frame of reference, i.e., $\boldsymbol{\omega}_0 = \boldsymbol{\omega}_f - \boldsymbol{\omega} = \omega_0 \hat{x}'$, with ω_0 and \hat{x}' denoting a fixed motor speed and a unit vector along the x' direction, respectively. The hydrodynamic interaction between sphere and wall (with a gap d) can be represented by a resistive matrix computed by the above BIM. The interaction between flagellum and wall, however, is neglected here due to their typical large separations.

FIG. 3. New Fig. S10. Statistical tests for the slipping ratio Q as a function of the dimensionless gap size d/R . **a** Both a linear regression (for $d/R < 2$; green) and a nonlinear regression (for all d/R ; red) were applied to the experimental data (filled circles; averaged over every one second). The shaded areas correspond to the 95% confidence bounds of the regressions (red: nonlinear, green: linear). The result for the hydrodynamic model (dashed line) reaches the maximum ($Q = 0.25$) at $d/R = 0$. **b** Variation of Q with respect to gap size (d/R) groups. Box plots are based on mean values calculated over 1 s time intervals. Values are significantly different among gap size groups (one-way ANOVA, $F_{2,141} = 19.07$, $P < 0.0001$). Asterisk symbol denotes the values significantly different between gap intervals in pairwise comparisons following one-way ANOVA (**, $P < 0.01$; ****, $P < 0.0001$; n.s., $P > 0.05$).

FIG. 4. New Fig. s11. Histogram of rosette size in this study shows a range of radius R from 2 to 6 μm , with its distribution peaks around 4 μm .

FIG. 5. New Fig. S12. Scanning electron microscopy (SEM) images. **a** An image of a typical predivisional *C. crescentus* cell, showing the asymmetric cell division between a mother (or stalked) and a daughter (or swarmer) cells. The stalk grown on the stalked end may potentially attach to an adhesive mass of holdfast shared with other stalked cells to form a rosette. The flagellum grown on the daughter cell may potentially be employed by the rosette for self-propulsion. **b–d** Images of *C. crescentus* rosettes under our culture protocols, showing the radial arrangement of stalked cells attached to a core. Arrows indicate partially damaged flagella due to SEM preparation. Scale bars, 1 μm .

FIG. 6. New Fig. S13. Segmented motile regimes in rosette's long-term dispersal. **a** A long-term recording (for over 40 min) of the speed of an individual rosette in the image plane (u_{x-y}) exhibited discrete motile regimes (with $u_{x-y} \sim 1 \mu\text{m/s}$), interrupted by those immotile regimes with much reduced speed (with $u_{x-y} \sim 0.1 \mu\text{m/s}$). The duration of these continuous motile regimes serves as an indicator for the duration of a set of functioning motors Δt_f achieved through flagellated daughter cells. An extremely long motile regime (e.g., the high-speed zone near the center, highlighted by dashed arrows) potentially contains several discrete motile regimes (driven by multiple sets of motors) without noticeable gaps. **b** The probability distribution functions (PDF) of u_{x-y} in these two regimes (in **a**) signify their characteristic speeds, which are significantly different: $1.11 \pm 0.62 \mu\text{m/s}$ (motile) v.s. $0.21 \pm 0.16 \mu\text{m/s}$ (immotile).

Reviewers' comments:

Reviewer #1 (Remarks to the Author):

In my opinion, the authors have adequately addressed and clarified all issues that were brought up by the reviewers in their rebuttal letter. Thus, I warmly recommend publication of the revised manuscript in Communications Biology.

Reviewer #2 (Remarks to the Author):

The authors have sufficiently answered the queries of their reviewers.

Reviewer #3 (Remarks to the Author):

The authors have satisfactorily addressed the issues I had raised. The manuscript is now suitable for publication.

Reviewer #4 (Remarks to the Author):

Summary

I appreciate that the authors provided a thorough reply to our comments and made substantial improvements to the manuscript on the questions that were raised during the first round of reviews. Overall, I believe the manuscript strongly benefits from these modifications and is closer to a publishable version. To improve it further, I attach a list of comments in chronological order following the main text. Most comments are minor, but comments displayed in bold need addressing before publication. I still recommend the publication of this manuscript in Communications Biology once these comments are addressed.

Comments

Introduction

The introduction has been largely improved compared to the previous version and now provides a clear context for the study.

a binary aggregate can be formed between both sessile and motile cells due to, for instance, variant possessions of motile organelles due to asynchronous development in life cycle

Avoid repetition of "due".

In this intermediate regime, the motile compartment of the dispersal unit is in principle capable of carrying the entire aggregate and allocating resources. This is the point of the paper and presented here it sounds like it is a known fact: Either formulate it as a working hypothesis or cite reference.

Reading through the paragraph, it was originally unclear that stalks can adhere to other cells and not only to surfaces (classic *C. crescentus* image). If there is a chance the targeted readership may be confused as well, clarify at some point in the paragraph.

Results

Sessile-motile coexistence leads to active rosette motilities

50 μ m-thick fresh growth medium Unclear, reformulate.

Replace "solitary cell" by "swarmer/swarming solitary cell" in Fig. 1c.

The authors very nicely explicited the reason for having a N of 9 rosettes in graph 1c, instead of the 18 tracked rosettes. This should also appear in the caption of Fig. 1c.

A number of 71 speed measurements for ballistic speed is mentioned in the main text. Are these 71 measurements all from different rosettes trajectories or are different ballistic legs from the same rosette counted as separate measurements? This information should be made available in the Methods section or in SI as it impacts the confidence in the measurement.

While comparing the swarmer cell's diffusive behavior to the rosette's, one could expect a discrepancy due to the fact that the first one is a pure 2D diffusive behavior when the second is a 3D diffusive behavior projected in 2D. A comment about this fact would strengthen the text. In particular, if the displacement in z are deemed negligible - give the reasoning behind.

Solid surfaces promote rosettes' rolling

In Fig. 3a, the distribution of angles is already a little skewed towards $3\pi/2$ for larger gaps. This is reflected in the main text by "Beta was distributed nearly uniformly between 0 and 2π ", which is a vague formulation which needs reformulation. A formulation that purely compares the two situations (small vs large gap), highlighting that one is more polarized than the other, would be more rigorous. Additionally, in the light of Fig. 3b that displays a continuous variation of the u-w

ratio as a function of the normalized gap, it is likely that the polarization to $3\pi/2$ is a continuum as the rosette gets closer to the bottom wall. This interpretation could be presented in the main text. If the authors have the supporting data to plot a parameter that represents the angle polarization as a function of the normalized gap, I believe it could strengthen their analysis.

In the caption of figure 3b, the authors mention an exponential fit. There is a priori no reason to choose such a mathematical form and the authors nicely present an alternative fitting in the SI. This effort could be mentioned in the caption or in the main text to highlight the robustness of the analysis.

Motor reversals result in reorientations of rolling rosettes

In Fig. 4, "solitary cell" should be replaced with "solitary swarmer cell" (or any formulation chosen for Fig. 1).

In the rosette trajectory example in Fig. 4a, the CCW (red) legs of the trajectory are systematically longer than the CW (blue) legs. This is counter intuitive given that Fig. 4b shows that the motor duration in both regimes is on average the same. Therefore, either the displayed trajectory is not representative of the typical behavior and should be replaced; or there is a significant difference in displacement speed between the two regimes. Either way, an explicit comment to clarify the situation is desirable. Data should be added in support of the latter case if it is the realistic scenario.

This similarity in motor behaviors further indicates that rosettes are each powered by a single member cell.

The authors made their claims less extreme about the number of cells throughout the manuscript, which reflected better the indirect nature of the proofs provided. Here, I think that "are" should here be changed in "can be" to be consistent with this change.

Potential mechanisms (or biases) for single-motor powering of rosettes

The addition of this new paragraph brings up valuable points for the presented work and convincing responses to the reviewers comments. However, I recommend moving it to the beginning of the discussion section for clarity. Associated with a brief summary of the observations that support that rosettes can be powered by single motors, it would appear more understandable and could strengthen the claim significantly.

Discussion

A lot of future work is announced in the discussion. Please refer to SI in the main text for the data that has been added. Overall, the discussion should focus more on this publication results than on preliminary observations. As a guide, I would recommend that the authors discuss in priority the data they are showing in SI.

An incipient division-of-labor with little or no intercellular coordination

Remove or clarify this sentence. Nothing in the current work rules out any intercellular coordination. This sentence should therefore be removed or be more speculative.

SI

Fig. S6 should mention the number of datapoints N in the histogram.

RESPONSE TO REVIEWER# 4

Summary

I appreciate that the authors provided a thorough reply to our comments and made substantial improvements to the manuscript on the questions that were raised during the first round of reviews. Overall, I believe the manuscript strongly benefits from these modifications and is closer to a publishable version. To improve it further, I attach a list of comments in chronological order following the main text. Most comments are minor, but comments displayed in bold need addressing before publication. I still recommend the publication of this manuscript in *Communications Biology* once these comments are addressed.

We again appreciate the reviewer's comments, which we try addressing in the following, one at a time.

Comments

Introduction

The introduction has been largely improved compared to the previous version and now provides a clear context for the study.

We thank the reviewer for this kind comment.

a binary aggregate can be formed between both sessile and motile cells due to, for instance, variant possessions of motile organelles due to asynchronous development in life cycle

Avoid repetition of "due".

We have rewritten the sentence as "a binary aggregate can be formed between both sessile and motile cells due to, for instance, variant possessions of motile organelles associated with asynchronous development in life cycle"

In this intermediate regime, the motile compartment of the dispersal unit is in principle capable of carrying the entire aggregate and allocating resources. This is the point of the paper and presented here it sounds like it is a known fact: Either formulate it as a working hypothesis or cite reference.

We have rewritten the sentence as "In this intermediate regime, it is unclear whether the motile compartment of the dispersal is capable of carrying the entire aggregate and thus allocating resources."

Reading through the paragraph, it was originally unclear that stalks can adhere to other cells and not only to surfaces (classic *C. crescentus* image). If there is a chance the targeted readership may be confused as well, clarify at some point in the paragraph.

We have added a sentence "Each stalk possesses a polysaccharide holdfast at the distal end, which allows for adhesion to solid surfaces or the holdfast from another cell" and relevant references immediately after the introduction of the stalk.

Results

Sessile-motile coexistence leads to active rosette motilities

50 μm -thick fresh growth medium Unclear, reformulate.

We have reformulated the sentence as "These rosettes were suspended in the growth medium confined between two coverslips (with a distance $\sim 50 \mu\text{m}$) ..."

Replace "solitary cell" by "swarmer/swarming solitary cell" in Fig. 1c.

We revised the text as suggested.

The authors very nicely explicit the reason for having a N of 9 rosettes in graph 1c, instead of the 18 tracked rosettes. This should also appear in the caption of Fig. 1c.

We have added a sentence "The difference in sample size between (b) and (c) is due to the fact that not all trajectories in (b) are sufficiently long for MSD calculations (see Methods)" in the figure caption.

A number of 71 speed measurements for ballistic speed is mentioned in the main text. Are these 71 measurements all from different rosettes trajectories or are different ballistic legs from the same rosette counted as separate measurements? This information should be made available in the Methods section or in SI as it impacts the confidence in the measurement.

These measurements were performed on 71 different rosettes. All speed statistics are sampled for $\gtrsim 1$ s for individual swarmer cells and rosettes (at 160 Hz for swarmer cells and at 20 Hz for slower swimming rosettes). We have

now included this information in *SI*.

While comparing the swarmer cell's diffusive behavior to the rosette's, one could expect a discrepancy due to the fact that the first one is a pure 2D diffusive behavior when the second is a 3D diffusive behavior projected in 2D. A comment about this fact would strengthen the text. In particular, if the displacement in z are deemed negligible - give the reasoning behind.

Due to the relatively thin sample thickness ($\lesssim 50 \mu\text{m}$) and the relatively high swimming speed $\sim 50 \mu\text{m/s}$, the axial (z) movement of a swarmer cell is geometrically restricted to be quasi-2D. In addition, the z range of all swarmer cells that we recorded (for over 10 s) are within $10 \mu\text{m}$, presumably subjected to a hydrodynamic attraction from cover slip surfaces (A. P. Berke et al., *Physical Review Letters*, **101**:038102, 2008). We therefore concluded that a 2D MSD characterizes reasonably well the diffusive behaviors of the swarmer cell. We have now included this rationale in Methods.

Solid surfaces promote rosettes' rolling

In Fig. 3a, the distribution of angles is already a little skewed towards $3\pi/2$ for larger gaps. This is reflected in the main text by "Beta was distributed nearly uniformly between 0 and 2π ", which is a vague formulation which needs reformulation. A formulation that purely compares the two situations (small vs large gap), highlighting that one is more polarized than the other, would be more rigorous. Additionally, in the light of Fig. 3b that displays a continuous variation of the $u-w$ ratio as a function of the normalized gap, it is likely that the polarization to $3\pi/2$ is a continuum as the rosette gets closer to the bottom wall. This interpretation could be presented in the main text. If the authors have the supporting data to plot a parameter that represents the angle polarization as a function of the normalized gap, I believe it could strengthen their analysis.

We have reformulated the previous vague sentence as "the probability distribution β was slightly skewed toward $\beta \approx 3\pi/2$, if not uniform between 0 and 2π ."

We have followed the reviewer's suggestion and analyzed the polarity of the PDF of β for various gaps, which is included in *SI* as a new figure (new Fig. S11). We also show that the polarity decreases with increased gaps, agreeing with the reviewer's surmise. This result has been included in the revised Fig. 3b and referred to in the main text:

"We also characterized the probability distribution $\rho(\beta)$ for various gap d (see *SI*, Fig. S11) by its polarity, i.e., $P = 2(\rho_{\max} - \rho_{\min})/(\rho_{\max} + \rho_{\min})$. Such a polarity P decreases with increasing d , as shown in Fig. 3b(inset), consistent with a rolling movement induced by the bottom surface."

We would also note that there was a mistake in our previous Fig. 3a: the previous values of the d/R were indeed d without normalization (due to a bug in the plotting script). We have corrected these values, and updated the plots accordingly. We intentionally chose the same value of d/R for large gap, i.e., $d/R > 2$, due to the new results on polarity analysis. We have also verified that other plots and our conclusions were not affected by this mistake.

In the caption of figure 3b, the authors mention an exponential fit. There is a priori no reason to choose such a mathematical form and the authors nicely present an alternative fitting in the *SI*. This effort could be mentioned in the caption or in the main text to highlight the robustness of the analysis.

We have included the following sentence in the caption:

"Note that the observed ratios are higher than that expected from pure hydrodynamic interactions for $d/R < 1$ (dashed line), as further confirmed by the results from both linear regression and analysis of variance (ANOVA) (see *SI*, Fig. S10)."

Motor reversals result in reorientation of rolling rosettes

In Fig. 4, "solitary cell" should be replaced with "solitary swarmer cell" (or any formulation chosen for Fig. 1).

We revised the text as suggested.

In the rosette trajectory example in Fig. 4a, the CCW (red) legs of the trajectory are systematically longer than the CW (blue) legs. This is counter intuitive given that Fig. 4b shows that the motor duration in both regimes is on average the same. Therefore, either the displayed trajectory is not representative of the typical behavior and should be replaced; or there is a significant difference in displacement speed between the two regimes. Either way, an explicit comment to clarify the situation is desirable. Data should be added in support of the latter case if it is the realistic scenario.

We agreed that the example trajectory of rosette shown in the previous Fig. 4a does a poor job in illustrating the distribution of motor durations in both directions. We have now replaced that plot with a more representative one in

the revised Fig. 4a.

This similarity in motor behaviors further indicates that rosettes are each powered by a single member cell.

The authors made their claims less extreme about the number of cells throughout the manuscript, which reflected better the indirect nature of the proofs provided. Here, I think that "are" should here be changed in "can be" to be consistent with this change.

We have replaced "are" with "can be".

Potential mechanisms (or biases) for single-motor powering of rosettes

The addition of this new paragraph brings up valuable points for the presented work and convincing responses to the reviewers comments. However, I recommend moving it to the beginning of the discussion section for clarity. Associated with a brief summary of the observations that support that rosettes can be powered by single motors, it would appear more understandable and could strengthen the claim significantly.

We have followed the reviewer's suggestion and moved these paragraphs to Discussion, with a brief summary of the observations supporting the single-motor powering speculations.

Discussion

A lot of future work is announced in the discussion. Please refer to SI in the main text for the data that has been added. Overall, the discussion should focus more on this publication results than on preliminary observations. As a guide, I would recommend that the authors discuss in priority the data they are showing in SI.

We have removed a substantial amount of preliminary observations and future work, replaced by more discussion of *SI* data, as recommended by the reviewer.

An incipient division-of-labor with little or no intercellular coordination

Remove or clarify this sentence. Nothing in the current work rules out any intercellular coordination. This sentence should therefore be removed or be more speculative.

We have removed this phrase.

SI

Fig. S6 should mention the number of datapoints N in the histogram.

We have now provided number of datapoints in used in the histogram:

"All speed statistics are sampled for $\gtrsim 1$ s for individual swarmer cells and rosettes (at 160 Hz for swarmer cells and at 20 Hz for the slower-swimming rosettes). The numbers of data points are $N = 70629$ from $N = 119$ cells and $N = 5295$ from $N = 71$ rosettes."

FIG. 1. New Fig. S11.

FIG. 2. Revised Fig. 3.

FIG. 3. Revised Fig. 4.

REVIEWERS' COMMENTS:

Reviewer #4 (Remarks to the Author):

I have reviewed the authors' modification following the last round of comments and I believe the manuscript is now ready for publication.